# Structural insight into the stabilization of microtubules by taxanes

Andrea E Prota[1]*, Daniel Lucena-Agell[2]†, Yuntao Ma[3]†, Juan Estevez-Gallego[2], Shuo Li[3], Katja Bargsten[1]‡, Fernando Josa-Prado[2], Karl-Heinz Altmann[4], Natacha Gaillard[1], Shinji Kamimura[5], Tobias Mühlethaler[1], Federico Gago[6], Maria A Oliva[2], Michel O Steinmetz[1,7], Wei-Shuo Fang[3]*, J Fernando Díaz[2]*

[1]Laboratory of Biomolecular Research, Division of Biology and Chemistry, Paul Scherrer Institute, Villigen PSI, Switzerland; [2]Centro de Investigaciones Biológicas Margarita Salas, Consejo Superior de Investigaciones Científicas, Madrid, Spain; [3]State Key Laboratory of Bioactive Substances and Functions of Natural Medicines, Institute of Materia Medica, Chinese Academy of Medical Sciences & Peking Union Medical College, Beijing, China; [4]Department of Chemistry and Applied Biosciences, Institute of Pharmaceutical Sciences, ETH Zurich, Zürich, Switzerland; [5]Department of Biological Sciences, Faculty of Science and Engineering, Chuo University, Tokyo, Japan; [6]Department of Biomedical Sciences, University of Alcalá, Madrid, Spain; [7]University of Basel, Basel, Switzerland

*For correspondence:
andrea.prota@psi.ch (AEP);
wfang@imm.ac.cn (W-ShuoF);
fer@cib.csic.es (JFernandoD)

†These authors contributed equally to this work

Present address: ‡leadXpro AG, PARK innovAARE, Villigen, Switzerland

**Abstract** Paclitaxel (Taxol) is a taxane and a chemotherapeutic drug that stabilizes microtubules. While the interaction of paclitaxel with microtubules is well described, the lack of high-resolution structural information on a tubulin-taxane complex precludes a comprehensive description of the binding determinants that affect its mechanism of action. Here, we solved the crystal structure of baccatin III the core moiety of paclitaxel-tubulin complex at 1.9 Å resolution. Based on this information, we engineered taxanes with modified C13 side chains, solved their crystal structures in complex with tubulin, and analyzed their effects on microtubules (X-ray fiber diffraction), along with those of paclitaxel, docetaxel, and baccatin III. Further comparison of high-resolution structures and microtubules' diffractions with the apo forms and molecular dynamics approaches allowed us to understand the consequences of taxane binding to tubulin in solution and under assembled conditions. The results sheds light on three main mechanistic questions: (1) taxanes bind better to microtubules than to tubulin because tubulin assembly is linked to a βM-loopconformational reorganization (otherwise occludes the access to the taxane site) and, bulky C13 side chains preferentially recognize the assembled conformational state; (2) the occupancy of the taxane site has no influence on the straightness of tubulin protofilaments and; (3) longitudinal expansion of the microtubule lattices arises from the accommodation of the taxane core within the site, a process that is no related to the microtubule stabilization (baccatin III is biochemically inactive). In conclusion, our combined experimental and computational approach allowed us to describe the tubulin-taxane interaction in atomic detail and assess the structural determinants for binding.

## Editor's evaluation

Here Prota et al. compare the action of the microtubule-stabilizing agent, taxol, with that of closely-related analogues and, as a result, successfully dissect the interactions and roles of different regions of the taxol molecule. The overall story is solid, providing new molecular insights, including defining and separating the lattice expansion effect from the lattice stabilization effect upon taxane binding.

This important work will be of interest to the microtubule cytoskeleton and structural biology communities.

## Introduction

The taxane paclitaxel is a drug included in the World Health Organization's List of Essential Medicines (*World Health Organization, 2021*). Taxanes, either alone or in combination with other

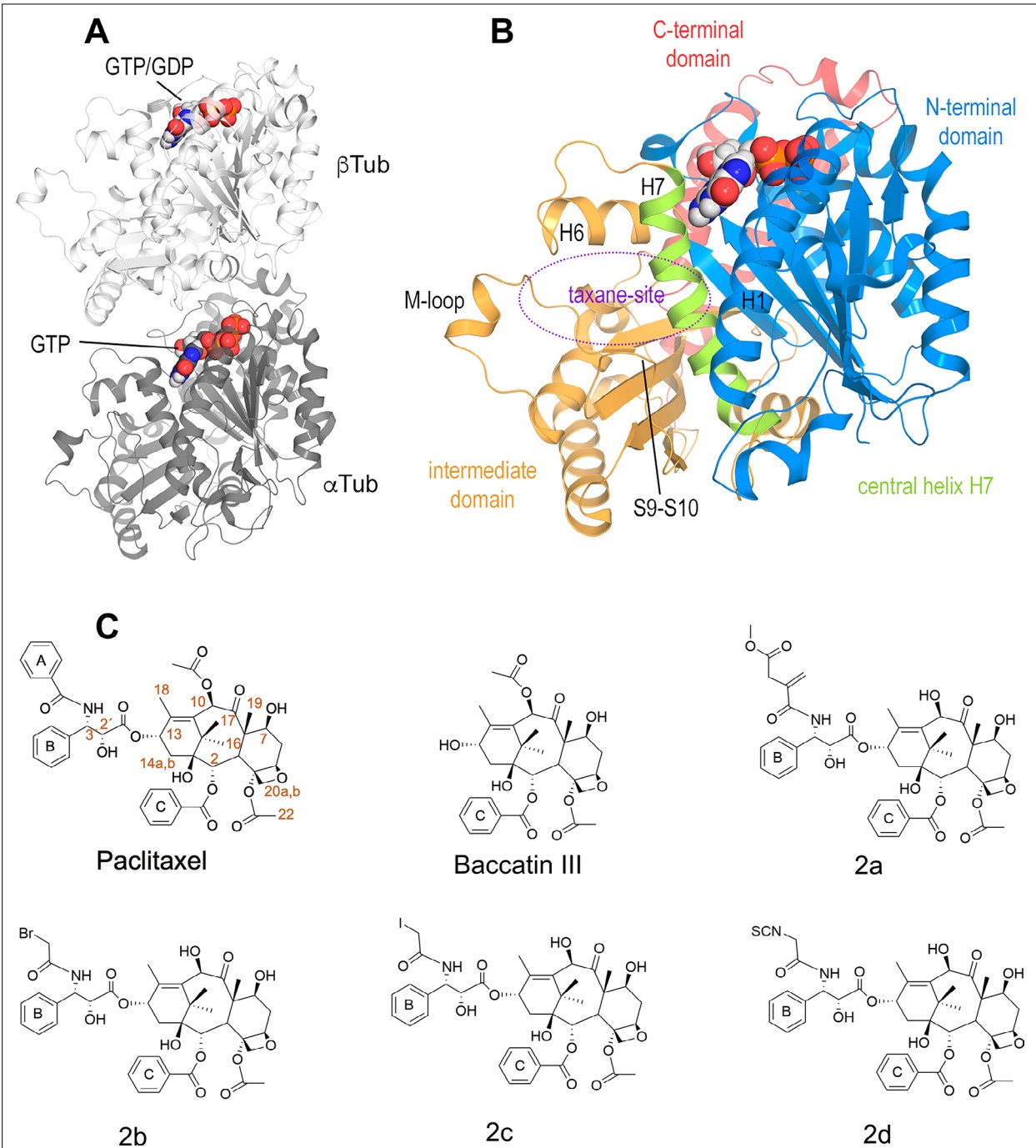

**Figure 1.** Structures of tubulin and ligands employed in the work. (**A**) Tubulin heterodimer (α-tubulin in gray and β-tubulin in white) in ribbon representation, where nucleotide binding sites have been highlighted in sphere representation (**B**) Structural features of the tubulin β-subunit. (**C**) Structures of taxanes used in this study.

chemotherapeutic agents, are important drugs for the treatment of several solid tumors, such as ovarian, lung, and breast cancer, as well as advanced Kaposi's sarcoma (*Ettinger, 1993*; *Arbuck et al., 1994*; *Saville et al., 1995*; *Lindemann et al., 2012*). The three taxanes in clinical use, paclitaxel (Taxol), docetaxel (Taxotere), and cabazitaxel (Jevtana), are part of a large family of chemically diverse compounds that bind to the so-called 'taxane site' of the αβ-tubulin heterodimer (*Field et al., 2013*; *Steinmetz and Prota, 2018*; *Figure 1A and B*), the building block of microtubules. However, the appearance of peripheral sensory neuropathy and other side effects caused by taxanes compromises treatment efficacy in the long term (*Gornstein and Schwarz, 2014*). Thus, understanding the underlying mechanism of microtubule stabilization by this class of antitubulin agents is an important requirement for future and safer drug development efforts.

Because taxane-site ligands stabilize microtubules and suppress their dynamics, they are collectively called microtubule-stabilizing agents. Several structures of microtubules in complex with taxane-site agents have been recently analyzed and solved by cryo-electron microscopy (cryo-EM) to resolutions ranging between ~3 and ~10 Å. For paclitaxel, it was initially suggested that the drug acts on longitudinal tubulin contacts along protofilaments in microtubules by allosterically expanding the microtubule lattice in the direction of its long filament axis (*Vale et al., 1994*; *Arnal and Wade, 1995*; *Alushin et al., 2014*), a notion that is also consistent with X-ray fiber diffraction data (*Estévez-Gallego et al., 2020*). However, more recent studies suggest that paclitaxel enhances lattice flexibility and acts on lateral tubulin contacts between protofilaments in microtubules through interactions with the M-loop of the β-tubulin subunit (βM loop) (*Kellogg et al., 2017*; *Debs et al., 2020*; *Manka and Moores, 2018*).

Besides directly acting on microtubules, taxane-site ligands also have the capacity to bind to unassembled tubulin dimers and promote their assembly into microtubules (*Schiff and Horwitz, 1981*; *Carlier and Pantaloni, 1983*; *Howard and Timasheff, 1988*; *Díaz et al., 1993*; *Buey et al., 2004*). Several structures of non-taxane agents bound to the taxane site of tubulin have been solved to resolutions ranging from 2.4 to 1.8Å by X-ray crystallography (*Prota et al., 2013a*; *Trigili et al., 2016*; *Prota et al., 2017*; *Balaguer et al., 2019*). These data suggested that one mode of action of some taxane-site ligands such as zampanolide (PDB ID 4I4T) or epothilone A (PDB ID 4I5O) on unassembled tubulin is to stabilize lateral tubulin contacts between protofilaments within microtubules by structuring and stabilizing the βM loop into a short α-helix (*Prota et al., 2013a*). In contrast, the absence of a helical structure for this segment in the presence of the taxane-site ligands dictyostatin (PDB ID 4MF4) and discodermolide (PDB ID 5LXT) (*Trigili et al., 2016*; *Prota et al., 2017*) suggests a different, still poorly understood mechanism of microtubule stabilization for these two classes of non-taxane agents.

In the case of taxanes, one hypothesis is that they preferentially bind to a specific conformation of tubulin. It is well established that tubulin displays two prominent conformations that are related to its assembly state (reviewed in *Knossow et al., 2020*): a 'straight' conformation present in assembled microtubules (denoted 'straight tubulin' hereafter) and a 'curved' conformation observed in unassembled tubulin (denoted 'curved tubulin' hereafter). The 'curved-to-straight' conformational transition is required for the formation of lateral tubulin contacts between protofilaments in the main shaft of microtubules. Some data suggest that the activation mechanism of taxanes facilitates the curved-to-straight conformational transition by preferentially binding to the straight conformation of tubulin (*Nogales et al., 1998*; *Elie-Caille et al., 2007*; *Benoit et al., 2018*).

Structural information of a taxane in complex with unassembled tubulin is currently unavailable. With the aim of providing insight into the mechanism of action of this important class of anticancer drugs and into the tubulin-taxane interaction, we solved the high-resolution structures of three different taxanes bound to curved tubulin by X-ray crystallography. We further analyzed the effects of different taxanes on the microtubule lattice by X-ray fiber diffraction. These studies were complemented with molecular dynamics (MD) simulations that shed light on issues that were not amenable to experimental verification. Taken together, our results suggest that the main reason for the differential affinity of taxane-site ligands for assembled tubulin and unassembled tubulin arises from two terms. First, the stabilization of the βM loop in an 'out' conformation compatible with the formation of specific lateral contacts in microtubules and second, the selectivity of the bulky C13 side chain for the assembled, straight conformational state of tubulin. Finally, we found that the occupancy of the taxane site results in a displacement of the S9-S10 loop of β-tubulin that

accounts for the observed microtubule expansion with no influence, however, on the straightness of tubulin protofilaments.

## Results

### High-resolution crystal structure of a tubulin-taxane complex

To determine the high-resolution structure of a taxane bound to curved tubulin, we performed both soaking and co-crystallization experiments using the previously described protein complexes termed $T_2R$-TTL and TD1. The former complex is composed of two αβ-tubulin heterodimers bound head-to-tail, the stathmin-like protein RB3, and the tubulin tyrosine ligase (PDB ID 4IIJ) (*Prota et al., 2013a*; *Prota et al., 2013b*); the latter complex contains one αβ-tubulin heterodimer and the DARP in D1 (PDB ID 4DRX) (*Pecqueur et al., 2012*). We did not succeed in procuring any valuable structural information from these two crystal ensembles using a first series of taxanes comprising paclitaxel, docetaxel, the more soluble 3'-*N-m*-aminobenzamido-3'-*N*-debenzamidopaclitaxel (N-AB-PT) (*Li et al., 2000*), and the engineered, high-affinity taxanes Chitax 40 (*Matesanz et al., 2008*) and Chitax 68 (*Ma et al., 2018*). We thus decided to approach the issue from a different angle and started off with baccatin III, a precursor in the biosynthesis of paclitaxel that contains both the C2-benzoyloxy ring C and the C10 acetate ester, but lacks the C13 side chain with both the 3'-N-benzamido phenyl ring A and the 3'-phenyl ring B moieties (*Samaranayake et al., 1993*; *Figure 1C*). Notably, baccatin III is largely biologically inactive despite displaying micromolar affinity for microtubules (*Parness et al., 1982*; *Lataste et al., 1984*; *Kingston, 2000*; *Andreu and Barasoain, 2001*).

We found that baccatin III shows detectable affinity ($K_b$ 25°C $3.0\pm0.5 \times 10^3$ M$^{-1}$) to unassembled tubulin, which is in the same range as for other compounds that have been co-crystallized with tubulin, such as epothilone A $8\pm3 \times 10^3$ M$^{-1}$ (*Canales et al., 2014*) and discodermolide $2.0\pm0.7 \times 10^4$ M$^{-1}$ (*Canales et al., 2011*). Therefore, we hypothesized that the presence of the C13 side chain of the aforementioned taxanes might preclude the binding to the curved tubulin form present in both the $T_2R$-TTL and the TD1 complexes. Subsequently, we succeeded in obtaining a $T_2R$-TTL-baccatin III complex structure that was solved at 1.9 Å resolution (PDB ID 8BDE) (*Figure 2A and D*; *Table 1*). We found that the ligand binds to the taxane site of curved tubulin with its C2-benzoyloxy ring C stacked between the side chains of βH229 and βL275 in the leucine-rich β-tubulin pocket lined by the side chains of βC213, βL217, βL219, βD226, βH229, βL230, and βL275 (*Figures 3A and 4A*). Its carbonyl oxygen forms a weak hydrogen bond to the main chain amide of βR278. The C10 acetate is exposed to the solvent and, together with the C12 methyl, is within van der Waals distance to βG370 of the βS9-βS10 loop. Furthermore, the oxetane oxygen and the C13 hydroxyl accept hydrogen bonds from the main chain amide nitrogen of βT276 and the βH229 imidazole NE2, respectively. The C4 acetate is buried in the hydrophobic pocket made up by βL230, βA233, βF272, βP274, βL275, βM302, βL371, and the aliphatic portion of the βR369 side chain.

### Generation of paclitaxel analogs that bind to tubulin crystals

Aiming to understand the implication on tubulin activation of the paclitaxel's bulky and hydrophobic C13 ring A moiety (or its equivalent *tert*-butyl in docetaxel) and to elucidate the reason why it apparently precludes binding to $T_2R$-TTL and TD1 crystals (see above), we devoted a synthetic effort to obtaining new taxane ligands with modified C13 side chains. We produced a series of modified taxanes bearing smaller groups than paclitaxel at the 3'-*N* position, namely, acrylamide **2a**, haloacetamides **2b**, and **2c**, and isothiocyanate **2d** (*Figure 1C*). We could measure binding of **2a** to unassembled tubulin dimers ($K_{b25°C}$ $0.8\pm0.3 \times 10^3$ M$^{-1}$), but not of N-AB-PT (*Li et al., 2000*), Chitax 40 (*Matesanz et al., 2008*), or Chitax 68 (*Ma et al., 2018*), thus indicating that the modification of the paclitaxel structure increased the binding affinity for unassembled tubulin. In fact (*Figure 2B, C, E and F*), we found unequivocal difference electron densities at the taxane site of β-tubulin in $T_2R$-TTL crystals soaked with **2a** (PDB ID 8BDF) and **2b** (PDB ID 8BDG) and refined the corresponding structures to 1.95 and 2.35 Å resolution, respectively (*Table 1*).

Interestingly, the electron densities of compounds **2a** and **2b** displayed a continuity between the 3'-N-attached moieties of both ligands and the side chain of His 229 of β-tubulin (βH229), suggesting the possible formation of a covalent adduct. For further validation, we collected additional X-ray diffraction data on $T_2R$-TTL crystals soaked with the haloacetamide derivative **2b** at the bromine peak

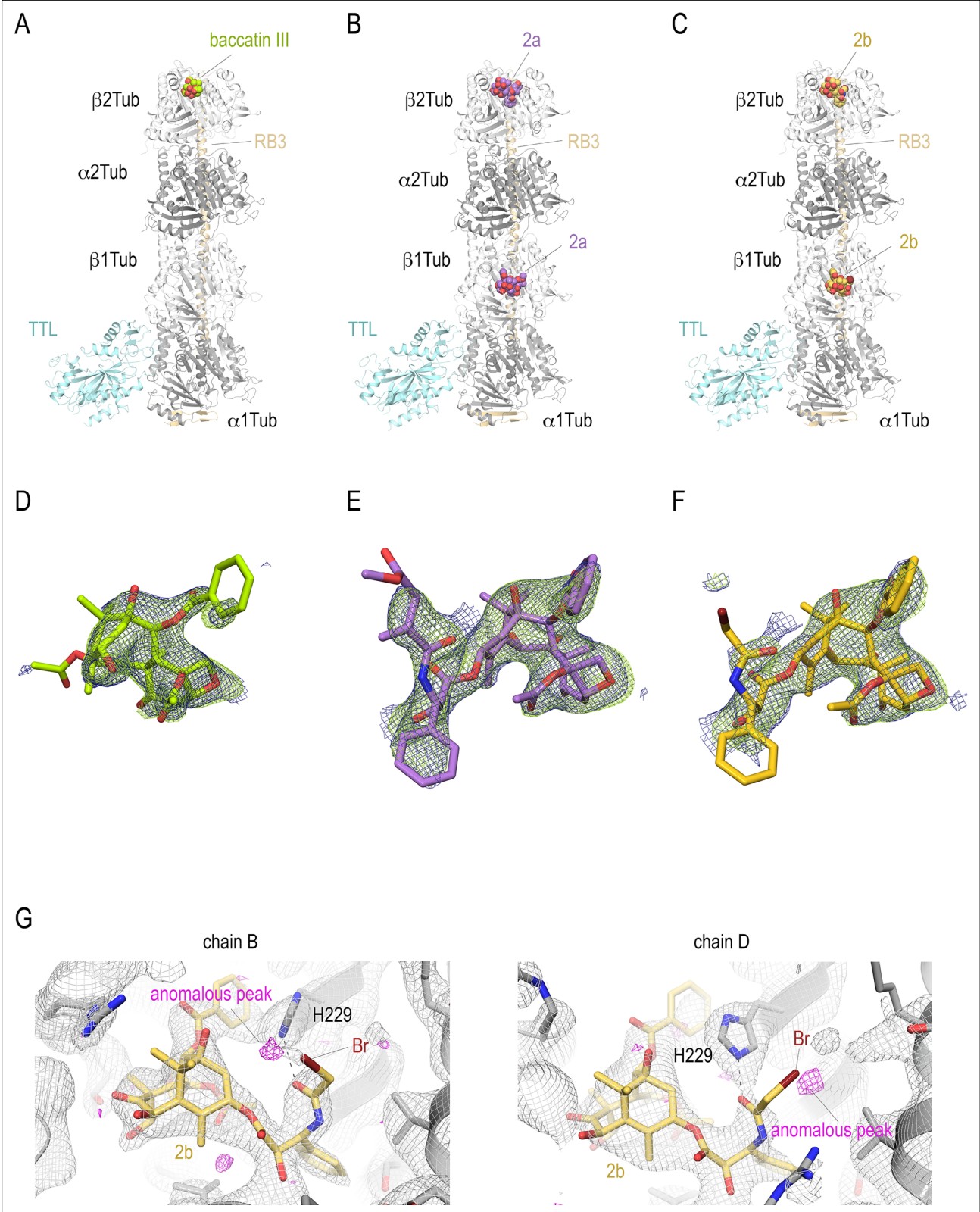

**Figure 2.** T₂R-TTL structures in complex with baccatin III, **2a**, and **2b**. Overall view of the T₂R-TTL-baccatin III (PDB ID 8BDE) (**A**), the T₂R-TTL-2a (PDB ID 8BDF) (**B**), and the T₂R-TTL-2b (PDB ID 8BDG) crystal structures. The α- and β-tubulin chains are colored in dark and light gray, respectively. The TTL chains (cyan) and the RB3 (yellow-orange) are shown in ribbon representation. The tubulin-bound ligands are displayed as spheres and are colored following the same color scheme as in the main figures. (**D–F**) Electron-density maps highlighting the bound baccatin III, **2a**, and **2b**. The SigmaA-

*Figure 2 continued on next page*

*Figure 2 continued*

weighted 2mFo − DFc (dark blue mesh) and mFo − DFc (light green mesh) omit maps are contoured at +1.0σ and +3.0σ, respectively. The map calculations excluded the atoms of the corresponding ligands. (**G**) Anomalous density peaks detected in both the binding sites in chains B and D of $T_2$R-TTL for the bromine moiety of compound 2b.

**Table 1.** X-ray data collection and refinement statistics.

| | $T_2$R-TTL-BacIII | $T_2$R-TTL-2a | $T_2$R-TTL-2b |
|---|---|---|---|
| **Data collection** | | | |
| Space group | $P2_12_12_1$ | $P2_12_12_1$ | $P2_12_12_1$ |
| Cell dimensions | | | |
| $a$, $b$, $c$ (Å) | 104.1, 157.2, 179.2 | 104.8, 157.9, 179.1 | 105.3, 158.6, 179.2 |
| Resolution (Å) | 49.2–1.9 (1.95–1.90) | 49.3–1.95 (2.00–1.95) | 49.4–2.35 (2.41–2.35) |
| $R_{merge}$ (%) | 10.7 (491.9) | 13.3 (516.6) | 17.4 (403.5) |
| $R_{meas}$ (%) | 11.1 (513.1) | 13.6 (526.1) | 17.7 (410.8) |
| $R_{pim}$ (%) | 3.3 (147.5) | 2.9 (102.9) | 2.6 (57.7) |
| $I/\sigma I$ | 16.5 (0.5) | 20.1 (0.7) | 20.1 (0.9) |
| CC half | 100 (17.8) | 100 (31.4) | 99.9 (46.6) |
| Completeness (%) | 100 (99.8) | 100 (100) | 100 (100) |
| Redundancy | 13.5 (12.4) | 27.3 (27.8) | 28.5 (28.3) |
| **Refinement** | | | |
| Resolution (Å) | 49.2–1.9 | 49.3–1.95 | 49.4–2.35 |
| No. unique reflections | 229654 | 215774 | 125168 |
| $R_{work}$/$R_{free}$ | 19.2/21.8 | 18.9/21.6 | 18.3/21.4 |
| No. atoms | | | |
| Protein | 17555 | 17389 | 17227 |
| Ligand | 42 | 120 | |
| Water | 861 | 883 | 166 |
| Average $B$-factors (Å²) | | | |
| Protein | 59.0 | 62.9 | 76.1 |
| Ligand (chain B/D) | n.a. / 109.2 | 111.4/102.8 | 146.6/144.9 |
| Water | 56.2 | 60.3 | 59.4 |
| Wilson $B$-factor | 41.7 | 43.1 | 56.9 |
| R.m.s. deviations | | | |
| Bond lengths (Å) | 0.003 | 0.003 | 0.002 |
| Bond angles (°) | 0.642 | 0.655 | 0.550 |
| Ramachandran statistics | | | |
| Favored regions (%) | 98.1 | 98.1 | 98.0 |
| Allowed regions (%) | 1.8 | 1.8 | 2.0 |
| Outliers (%) | 0.1 | 0.1 | 0 |

For each structure, data were collected from a single crystal. Values in parentheses are for highest-resolution shell.

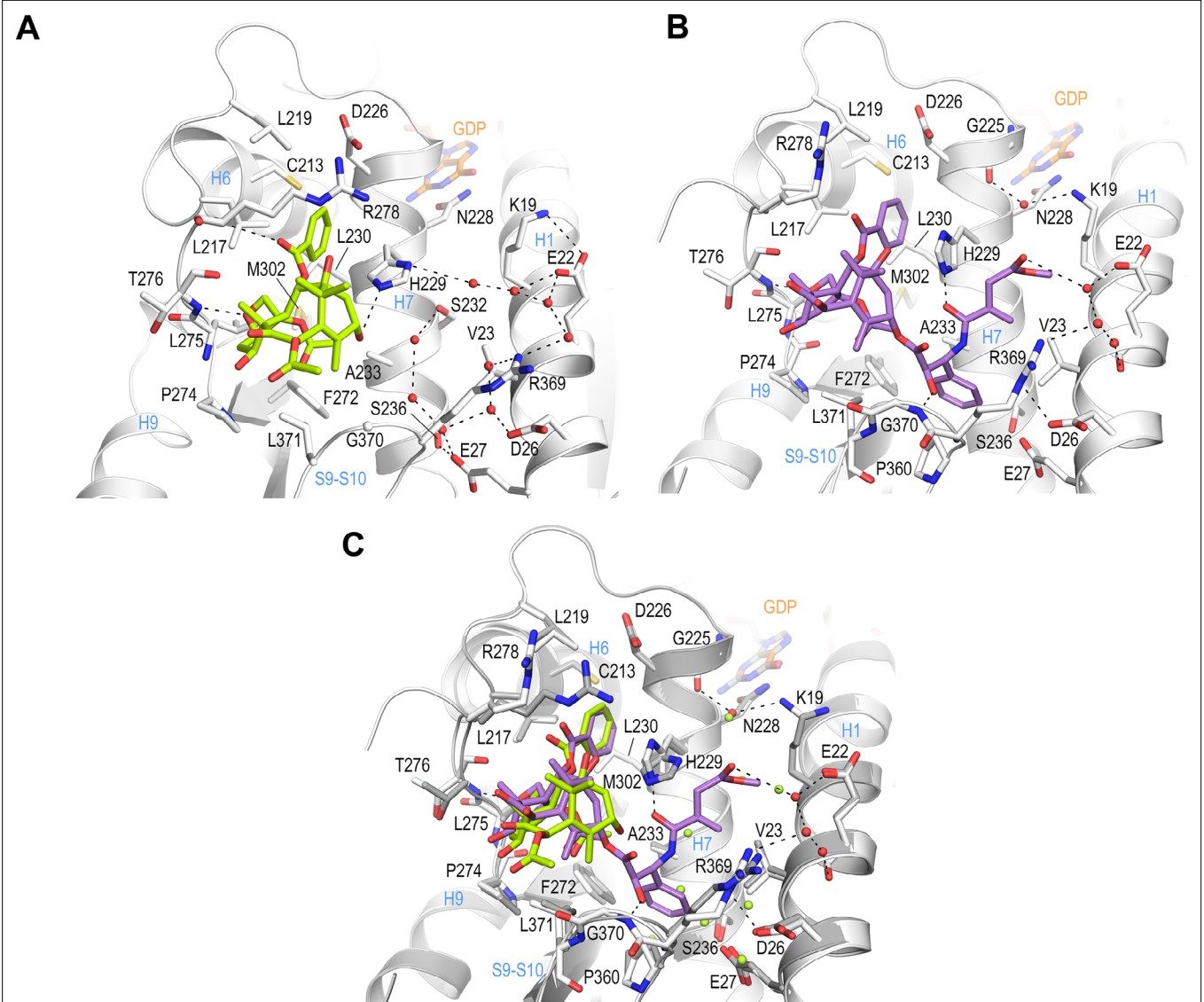

**Figure 3.** Crystal structure of T$_2$R-TTL-baccatin III (PDB ID 8BDE) and T$_2$R-TTL-2a (PDB ID 8BDF) complexes. (**A**) Close-up view of the interaction network observed between baccatin III (lemon) and β-tubulin (light gray). Interacting residues of tubulin are shown in stick representation and are labeled. Oxygen and nitrogen atoms are colored red and blue, respectively; carbon atoms are in lemon (baccatin III) or light gray (tubulin). Hydrogen bonds are depicted as black dashed lines. Secondary structural elements of tubulin are labeled in blue. (**B**) Close-up view of the interaction of **2a** (violet) with β-tubulin in the same view and representation as in (**A**). (**C**) The same close-up view as in (**A**) and (**B**) with the superimposed baccatin III (lemon) and **2a** (violet) complex structures. Water molecules belonging to the baccatin III structure are represented as lemon spheres.

wavelength of 0.91501 Å. After rigid body and restrained refinement, we detected two clear anomalous difference peaks in electron densities at the taxane sites of the two tubulin dimers in the T$_2$R-TTL crystals soaked with **2b** (*Figure 2G*), which did not support covalent bond formation. Furthermore, refinement cycles performed in parallel with **2a** modeled in both the covalent and the non-covalent form, resulted in clear electron density for the non-covalent model, while red difference peaks for the covalent form were always present after refinement (not shown). Accordingly, we interpreted the continuous electron density observed in the T$_2$R-TTL-**2a** structure as a strong hydrogen bond between the βH229 NE2 and the C39 carbonyl of the ligand side chain rather than a covalent bond (*Figure 3B*).

The T$_2$R-TTL-**2a** complex structure revealed that **2a** engages in comparable interactions to curved tubulin by means of both its C2-benzoyloxy ring C and its oxetane moieties, as found for baccatin III (*Figure 3A and B*). However, the core ring system of **2a** is tilted toward helix βH6 and strand βS7 by ~20° (angle between the two C1-C9 axes; rmsd$_{bacIII-2a}$ of 0.794 Å for 39 core atoms), thereby

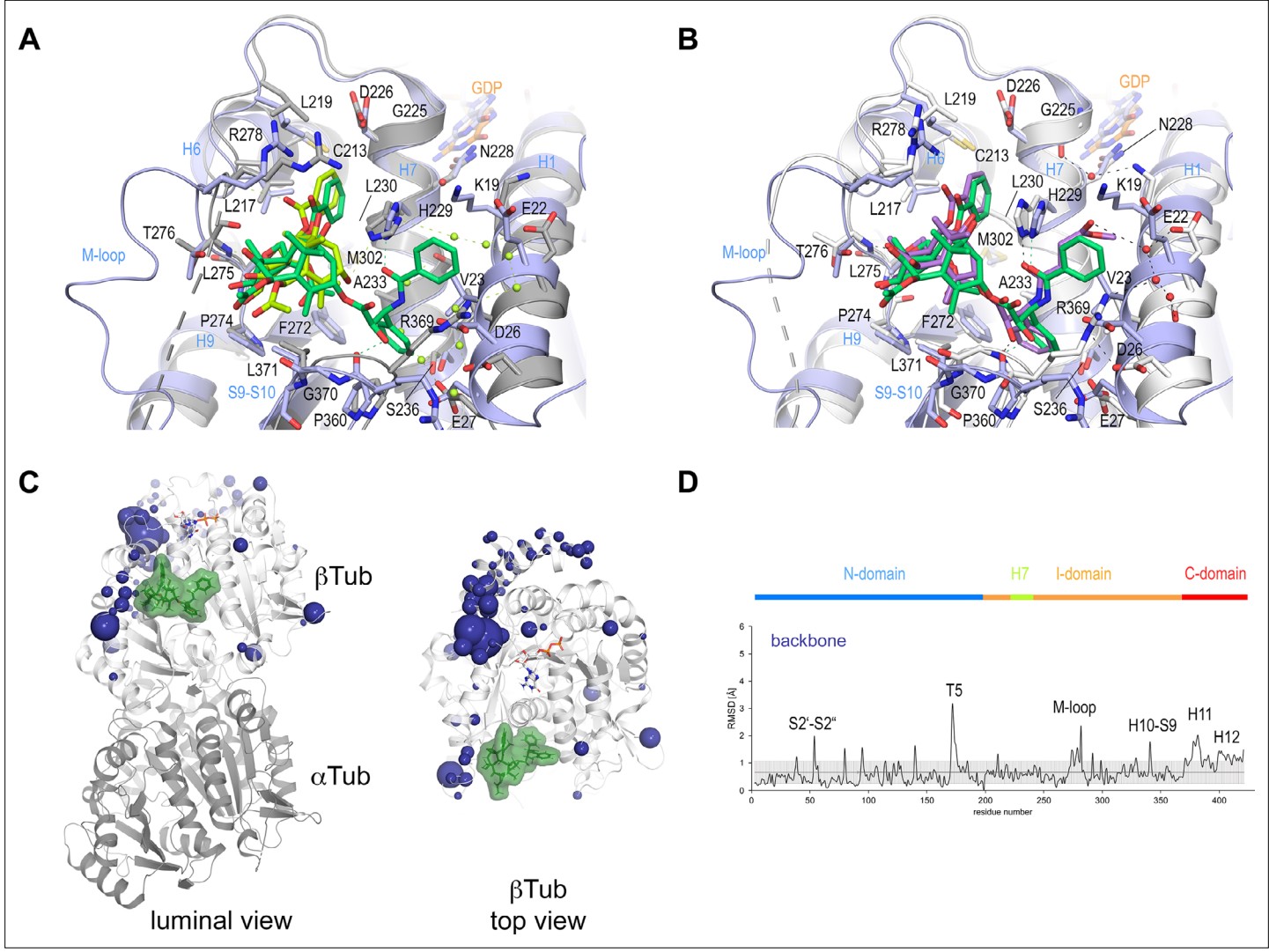

**Figure 4.** Comparison of taxane binding to unassembled curved versus assembled straight tubulin. (**A**) Close-up view of the superimposed baccatin III bound (ligand in lemon; protein in gray ribbon and sticks) to curved tubulin (PDB ID 8BDE) and paclitaxel bound to straight tubulin as found in a microtubule (PDB ID 6WVR; ligand in dark green; protein in slate ribbon and sticks) structures. Interacting residues of tubulin are shown in stick representation and are labeled. Oxygen and nitrogen atoms are colored red and blue, respectively. Hydrogen bonds are depicted as black dashed lines. Secondary structural elements of tubulin are labeled in blue. Water molecules belonging to the baccatin III structure are represented as lemon spheres. The structures were superimposed onto their taxane sites (residues 208–219+225–237+272–276+286–296+318–320+359–376); root-mean-square deviations (rmsd) 0.894 Å (52 C$_\alpha$ atoms). (**B**) Close-up view of superimposed **2a** bound to curved tubulin (PDB ID 8BDF) (ligand in violet; protein in gray ribbon and sticks) and paclitaxel bound to straight tubulin (PDB ID 6WVR; ligand in dark green; protein in slate ribbon and sticks) structures (rmsd 0.826 Å over 52 C$_\alpha$ atoms) using the same settings as in (**A**). (**C**) Conformational changes on β-tubulin induced by paclitaxel upon binding to straight tubulin in microtubules (PDB ID 6WVR). The α-tubulin and β-tubulin chains are in ribbon representation and are colored in dark and light gray, respectively. The rmsd differences between unbound and paclitaxel-bound straight tubulin are represented as dark (backbone rmsd) blue spheres. Only the rmsd differences above a threshold of average ± standard deviation are displayed. The sphere radii correspond to the average-subtracted rmsd values displayed in panel (**D**). (**D**) Rmsd plots of backbone positions between the paclitaxel bound (PDB ID 6WVR) and the apo (PDB ID 6DPV) straight tubulin in microtubules. The gray error bar represents the average rmsd ± standard deviation. The top bar is colored according to the following domain assignment: N-terminal domain (N-domain., marine blue), intermediate domain (I-domain, orange), central helix βH7 (lemon), and C-terminal domain (C-domain, red). The β-tubulin chains of the corresponding structures were superimposed onto their β-tubulin N-terminal β-sheets (rmsd 0.304 Å over 30 C$_\alpha$).

adopting a pose that is closer to that observed for paclitaxel bound to straight tubulin in microtubules (PDB ID 6WVR; rmsd$_{2a\text{-paclitaxel}}$ of 0.845 Å for 56 core atoms; rmsd$_{bacIII\text{-paclitaxel}}$ of 1.048 Å for 42 core atoms; *Figure 4B*).

Similar to paclitaxel bound to straight tubulin, the C39 carbonyl of the C13-3'-*N*-acrylamide moiety of **2a** forms a hydrogen bond to the βH229 NE2 in curved tubulin (*Figures 3B and 4B*). The terminal ester moiety of **2a** is exposed to the solvent and it forms water-mediated hydrogen bonds to the side chains of βE22 and βR369 of β-tubulin; it lodges within a space that is otherwise occupied by crystallographic water molecules in the curved tubulin-baccatin III structure. This favorable hydrogen bond network cannot be established by the 3'-*N*-benzamido phenyl ring A of paclitaxel in the curved tubulin conformation. Moreover, both the water molecules and the hydrophobic portions of the Lys19, Glu22, and Val23 side chains on helix H1 are too far apart for being able to provide favorable forces to stabilize the ring A. In the context of paclitaxel-bound microtubules (straight tubulin), the helix H1 moves closer toward helix H7, thereby allowing these three side chains to form a hydrophobic cavity that stabilizes the A ring, which suggest a structural mechanism for the higher affinity of paclitaxel observed for the straight tubulin conformation. Moreover, the helix H1 movement causes the side chain of βD26 to occupy the space of the βR369 side chain, which adopts a flipped-out conformation. This arrangement provides additional stabilization through a polar interaction to the 3' amide nitrogen of paclitaxel and supports a more favorable binding of paclitaxel to microtubules (*Figures 3B and 4B*). The absence of the C10 acetate in **2a** relative to baccatin III has little impact on the conformation of the secondary structural elements that shape the taxane site (*Figure 3C*).

Together, these structural data provide – for the first time – a high-resolution structural description of the interaction of taxanes harboring a C13 side chain with unassembled, curved tubulin. They indicate that the main interaction energy of this class of antitubulin agents is mediated by their common baccatin III core moieties. They further reveal that the taxane pose in both curved and straight tubulin is very similar; however, subtle structural details reveal why paclitaxel binds more favorably to straight tubulin. The knowledge of these structural determinants may support the development of next-generation taxanes to better tune their mechanism of action, thereby opening a new window to control undesired side effects. Overall, our results suggest that the tubulin-**2a** structure is an excellent model to study the interaction of paclitaxel with curved tubulin at high resolution and that X-ray crystallography is a valuable method to analyze the molecular mechanism of action of microtubule-stabilizing agents binding to the taxane site.

## Conformational changes upon taxane binding to curved and straight tubulin

Next, we investigated the conformational changes induced by binding of baccatin III and **2a** to curved tubulin. To this end, we first superimposed the crystal structures of apo tubulin (PDB ID 4I55), tubulin-baccatin III (PDB ID 8BDE), and tubulin-**2a** (PDB ID 8BDF) onto the N-terminal β-sheets of β-tubulin (residues 3–9, 63–66, 132–138, 163–169, and 198–202), and calculated the root-mean-square deviations (rmsd) between the apo and the two ligand-bound states (rmsd$_{BacIII}$ 0.08 Å of 29 C$_\alpha$; rmsd$_{2a}$ 0.10 Å of 29 C$_\alpha$). These rmsd values were also plotted and mapped onto the corresponding structures to highlight the major regions of conformational change.

As shown in *Figure 5*, significant and comparable conformational changes were observed for backbone atoms of the βT5 loop and the N-terminal segment of the βM loop in both the tubulin-baccatin III and tubulin-**2a** complex structures. Interestingly, the βT5 loop that is prominently involved in establishing longitudinal tubulin contacts along protofilaments is oriented in the active 'out' conformation in both structures (*Nawrotek et al., 2011*). This observation indicates an allosteric crosstalk between the taxane site and the βT5 loop possibly via the central helix βH7 and the guanosine nucleotide bound to β-tubulin. In the case of the βM loop, we only found well-defined electron densities for its N-terminal section up to residue βR278, while the remaining portion of the loop appeared disordered in both complex structures. This partial βM loop structuring has been observed previously in tubulin complexes with the taxane-site ligands dictyostatin and discodermolide (*Trigili et al., 2016*; *Prota et al., 2017*; note that the taxane-site ligands zampanolide and epothilone A promote the structuring of the βM loop into a helical conformation *Prota et al., 2013a*). A direct effect of taxanes on the βM loop is consistent with the notion that paclitaxel stabilizes this secondary structural element in two discrete conformations giving rise to heterogeneous lateral microtubule-lattice contacts (*Debs et al.,*

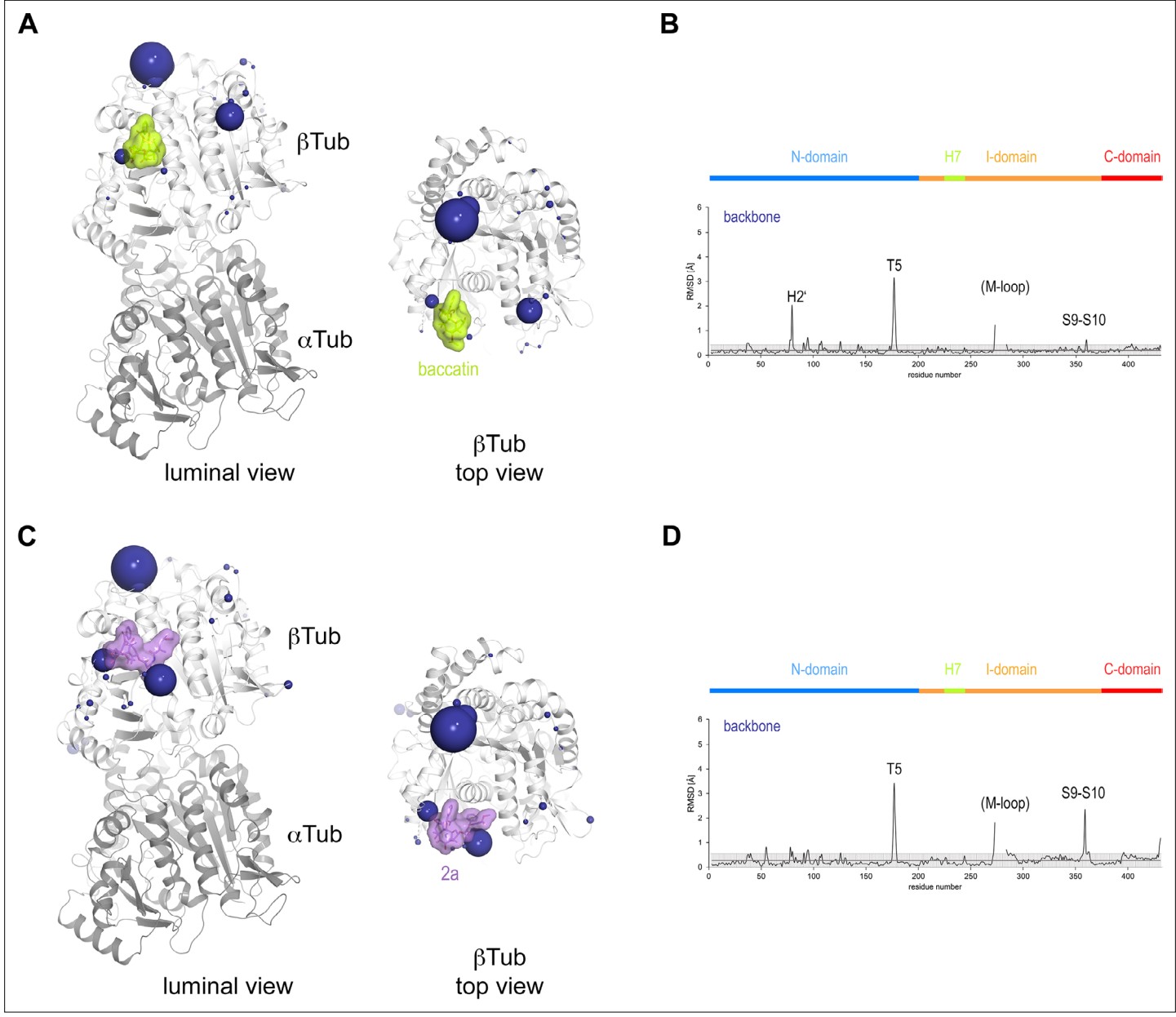

**Figure 5.** Conformational changes induced by taxane binding to unassembled, curved tubulin. (**A**) Conformational changes on the backbone atoms (dark blue) of the β-tubulin chain induced by baccatin III upon binding to curved tubulin. The tubulin chains are in ribbon representation and are colored in dark (α-tubulin) and light (β-tubulin) gray, respectively. The root-mean-square deviation (rmsd) values of the superimposed unbound and baccatin III-bound curved tubulin are represented as dark blue (backbone rmsd) spheres, respectively. Only the rmsd values above a threshold of average + standard deviation are displayed. The sphere radii correspond to the average-subtracted rmsd values displayed in panel (**B**). (**B**) Rmsd plots of the backbone (bottom) positions between the baccatin bound (PDB ID 8BDE) and the apo (PDB ID 4I55) curved tubulin state. The gray error bar represents the average rmsd ± standard deviation. The top bar is colored according to the following domain assignment: N-terminal domain (N-domain, marine blue), intermediate domain (I-domain, orange), central helix H7 (lemon), C-terminal domain (C-domain, red). The β-tubulin chains of the corresponding structures were superimposed onto their β-tubulin N-terminal β-sheet (rmsd 0.08 Å over 29 C$_\alpha$). (**C**) Conformational changes on the backbone atoms (dark blue) of the β-tubulin chain induced by **2a** upon binding to curved tubulin. (**D**) Rmsd plots of the backbone (bottom) positions between the **2a** bound (PDB ID 8BDF) and the apo (PDB ID 4I55) curved tubulin state (rmsd 0.10 Å over 29 C$_\alpha$). The same display settings as in (**B**) are applied.

*2020*). We also found significant conformational changes in the βS9-βS10 loop, which were more prominent in tubulin-**2a** than in tubulin-baccatin III. This finding can be explained by the presence of a C13 side chain in **2a** that needs more room for accommodation inside the taxane site compared to baccatin III, which lacks a C13 side chain. Finally, we observed a conformational change of the H2′ helix in the tubulin-baccatin III structure, which was absent in tubulin-**2a**.

To investigate the effect of the observed conformational changes on the relative domain arrangements in β-tubulin of the individual complexes, we further superimposed the β-tubulin chains of apo tubulin, tubulin-baccatin III, and tubulin-**2a** onto their central βH7 helices (residues 224–243). For tubulin-baccatin III, a subtle relative twist between the N-terminal and the intermediate domains was observed (*Figure 6*; *Videos 1 and 2*), while binding of **2a** rather caused both the N-terminal and intermediate domains of β-tubulin to move slightly apart (*Figure 6*; *Videos 3 and 4*). Thus, taxane binding to tubulin causes global, but subtle conformational rearrangements.

We next wondered whether similar conformational changes are also observed in straight tubulin in the context of a microtubule upon paclitaxel binding. To this end, we performed the same type of analysis by superimposing the N-terminal β-sheets of β-tubulin from the cryo-EM reconstruction of paclitaxel-bound guanosine diphosphate (GDP)-microtubules (PDB ID 6WVR) onto the corresponding domains of the undecorated apo GDP-microtubule structure (PDB ID 6DPV; rmsd 0.304 Å 30 $C_\alpha$). The rmsd analysis revealed similar significant conformational changes of both the βT5 and the βM loops as observed for the taxanes bound to curved tubulin, however, no prominent perturbations of the βS9-βS10 loop could be detected. In addition, we found significant conformational changes of the 'βS2-βS2'' loop (the secondary structural elements that interact with the βM loop of the neighboring protofilament) and the C-terminal βH11-βH12 helix region (*Figure 4C and D*), which were not detected in the curved tubulin structures.

Together, these results suggest that taxane binding in the context of the microtubule have an effect on the lateral contact established by the βM loop, an effect that cannot be detected in curved tubulin as this contact does not exist in the crystal. Moreover, we observe an activation effect on the T5 loop, but do not see any direct structural evidence for that. Therefore, taxane binding likely exerts this activation by affecting the dynamicity of helix H7, thereby establishing a crosstalk to the T5 loop through the nucleotide.

## Effects of taxanes on microtubule lattice parameters

We have previously validated X-ray fiber diffraction of shear-flow aligned microtubules as an accurate technique to determine microtubule lattice parameters (*Estévez-Gallego et al., 2020*; *Kamimura et al., 2016*). In such diffraction patterns, the meridional 4 nm layer line is related to the axial helical rise of tubulin monomers in the microtubule lattice. When the lattice is expanded in the direction of the helix axis, a second, weaker 8 nm layer line emerges due to the length difference between the α- and β-tubulin subunits (*Kamimura et al., 2016*) and the position of the 1 nm layer line corresponding to the fourth harmonic of the 4 nm layer line moves toward the center of the image.

We used this method to analyze the effect of different conditions on the microtubule lattice (*Figure 7*, *Table 2*). We first analyzed microtubules that were assembled in the presence of either guanosine triphosphate (GTP) (producing GDP-microtubules) or the slowly hydrolyzable GTP analogue GMPCPP (producing GMPCPP-microtubules) and found that the tubulin dimer rise increased by 0.24 nm (from 8.12±0.02 to 8.36±0.02 nm, respectively) in the presence of GMPCPP, a distance that is consistent with that found in previous studies. Concomitantly, the microtubule radius increased from 11.42±0.1 nm for GDP-microtubules to 11.63±0.1 nm for GMPCPP-microtubules, which translates into an increase of the average protofilament number (av. PF nr.) from 12.9 to 13.3, respectively. An increase in both tubulin dimer rise and number of protofilaments for GMPCPP-microtubules compared to GDP-microtubules has been reported previously (*Vale et al., 1994*; *Alushin et al., 2014*; *Hyman et al., 1992*; *Yajima et al., 2012*).

As shown in *Figure 7* and *Table 2*, and when compared to GDP-bound microtubules, both paclitaxel-bound microtubules and docetaxel-bound microtubules displayed a similar lattice expansion of 0.24 nm as seen for GMPCPP-bound microtubules. Interestingly, while paclitaxel-bound microtubules show a reduced microtubule radius of 10.97±0.1 nm (av. PF nr., 12.21), docetaxel-bound microtubules displayed a radius of 11.53±0.1 nm (av. PF nr., 12.9), which is similar to the value obtained for GDP-microtubules. In the case of paclitaxel, this expansion occurred either when the drug was added

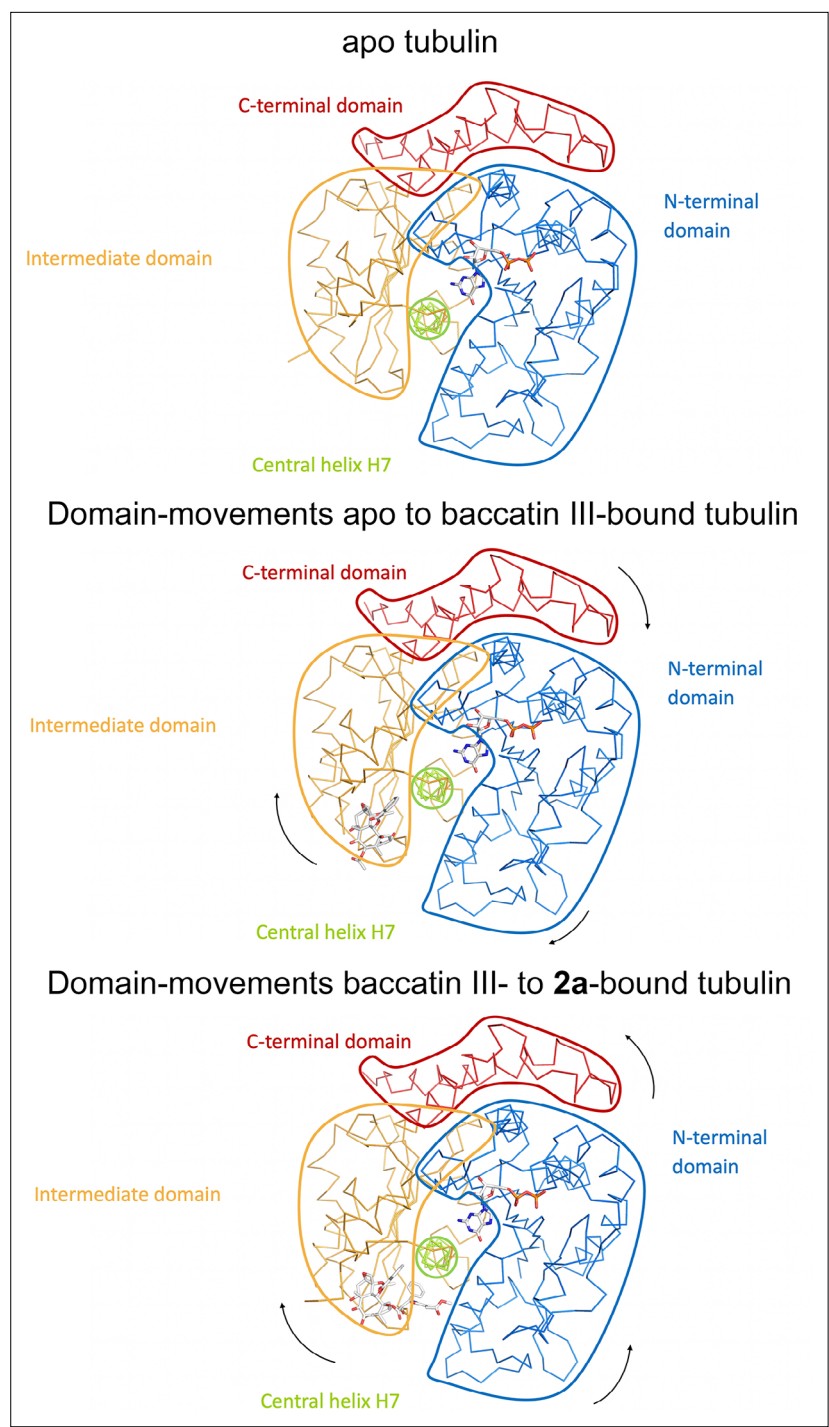

**Figure 6.** Schematic representation of subtle domain movements observed from apo to baccatin III- to 2a-bound curved tubulin. The three structures were superimposed onto their central helices βH7 to highlight better the subtle domain movements relative to each other. The individual domains are colored according to their domain assignment and their borders are contoured using the same color scheme: N-terminal domain (N-domain, marine blue), intermediate domain (I-domain, orange), central helix βH7 (lemon), C-terminal domain (C-domain, red). The directions of the individual movements are highlighted with black arrows.

before the polymerization reaction was started with GTP- or GDP-tubulin, or when it was added to preformed microtubules, in consonance with the rapid structural transitions of microtubules observed upon taxane addition (*Díaz et al., 1998*). Interestingly, microtubules with bound **2a**, **2b**, or baccatin III showed similar lattice expansion as those bound to paclitaxel or docetaxel. Note that the diffraction

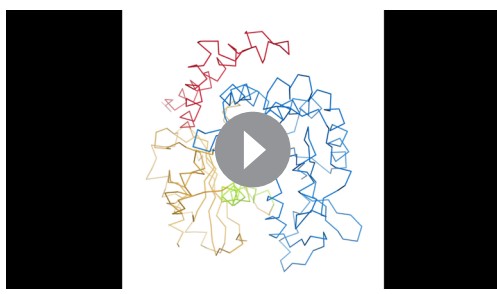

**Video 1.** Conformational transition from apo to baccatin III-bound, unassembled tubulin state. Top view on β-tubulin (onto the 'plus end' in the context of a microtubule).

https://elifesciences.org/articles/84791/figures#video1

patterns of microtubules stabilized with **2a** or **2b** showed a diffuse 1 nm layer line that reflects variations in the tubulin monomer (and consequently dimer) rise, in clear contrast to those bound by paclitaxel and docetaxel, which displayed a sharp band, that is, a robust monomer rise.

Taken together, these results suggest that taxanes with or without a C13 side chain have the capacity to expand the microtubule lattice and thus affect longitudinal tubulin contacts along protofilaments. They further indicate that the nature of the C13 side chain can affect the radius of a microtubule and thus lateral tubulin contacts between protofilaments. We note, however, that microtubules assembled in the presence of baccatin III, which lacks a C13 side chain, display the same radius as paclitaxel-bound microtubules.

Thus, the presence of a C13 side chain per se does not seem to modulate lateral tubulin contacts between protofilaments.

## MD simulation analysis of taxane binding to tubulin, protofilament, and microtubule lattice models

Although the high-resolution crystallographic structures discussed above provide detailed information of the taxane binding site for the ligands, no large differences were observed between apo- and taxane-bound tubulin structures, indicating that additional ligand effects may be related to the dynamic behavior of the protein. For these reasons and to gain further insight into the underlying mechanism of taxane-induced microtubule stabilization, we next used MD simulations to study the behavior of different tubulin assemblies in solution. To this end, we built three types of fully solvated molecular models representing the different oligomerization states of tubulin: (i) the αβ-tubulin heterodimer; (ii) a short protofilament consisting of three longitudinally concatenated tubulin dimers ((αβ-tubulin)$_3$); and (iii) a minimalist representation of a microtubule lattice (*Sánchez-Murcia et al., 2019*) made up of two laterally associated protofilament fragments (($\alpha_1$-$\beta_1$-$\alpha_2$)/($\alpha_{1'}$-$\beta_{1'}$-$\alpha_{2'}$)). All models were created in their apo- and taxane-bound forms. Baccatin III, **2a**, and paclitaxel were chosen as representative taxane ligands for our fully atomistic simulations.

In good agreement with the previous structural results, the MD simulations of the αβ-tubulin heterodimer pointed to the βM loop as the most likely structural element responsible for the selective recognition of the microtubule-assembled tubulin form by taxanes. All taxane-site ligands, including paclitaxel, docetaxel (*Díaz et al., 1993*), discodermolide (*Canales et al., 2011*), epothilone A (*Canales et al., 2014*), and **2a** show a loss of affinity of at least four orders of magnitude when binding to unassembled tubulin relative to binding to microtubules while covalent binders like zampanolide react

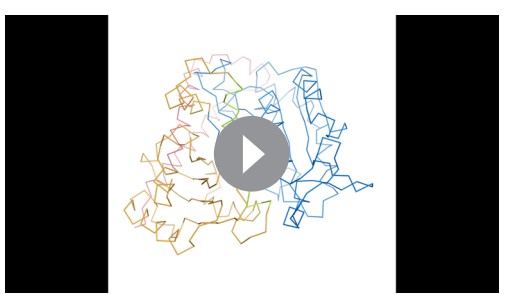

**Video 2.** Conformational transition from apo to baccatin III-bound, unassembled tubulin state. Luminal view on β-tubulin (view from the lumen in the context of a microtubule).

https://elifesciences.org/articles/84791/figures#video2

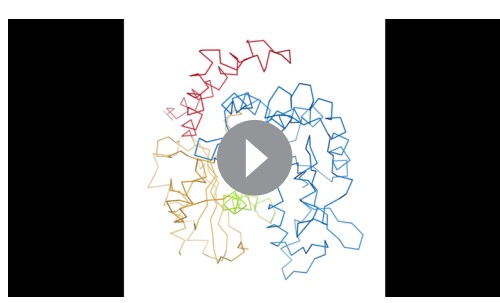

**Video 3.** Conformational transition from apo to 2a-bound, unassembled tubulin state. Top view on β-tubulin (onto the 'plus end' in the context of a microtubule).

https://elifesciences.org/articles/84791/figures#video3

**Figure 7.** Fiber diffraction patterns of microtubules. Microtubules assembled from guanosine triphosphate (GTP)-tubulin and paclitaxel (**A**), GTP-tubulin and docetaxel (**B**), GTP-tubulin and **2a** (**C**), GTP-tubulin and **2b** (**D**), and GTP-tubulin and baccatin III (**E**) are shown.

slowly with unassembled tubulin compared to microtubules (*Field et al., 2012*). Even baccatin III, which lacks the side chain altogether, has an affinity for the unassembled state that is still two orders of magnitude lower compared to the microtubule-assembled state ($3 \times 10^3$ M$^{-1}$ vs. $1.5 \times 10^5$ M$^{-1}$), a finding that is not explained by our crystallographic data. In our simulations of the tubulin dimer model, we found that the βM loop is the most flexible region (*Figure 8* top), in good accord with the fact that no density is usually observed for this β-tubulin element in most crystallographic structures. During the course of the MD simulations, this loop was not structured as an α-helix in any of the models studied; instead, it was found to assume a relatively stable, extended hairpin conformation that interacted with and blocked access to the taxane site. Even when the βM loop was initially modeled as an α-helix (as present in all microtubule structures solved by cryoelectron microscopy; *Alushin et al., 2014*; *Kellogg et al., 2017*; *Debs et al., 2020*; *Manka and Moores, 2018*), this secondary structure element was rapidly lost during the simulated trajectory (*Figure 8*, bottom) regardless of whether or not baccatin III, **2a**, or paclitaxel was bound at the taxane site (*Figure 9AB*). One likely reason for this behavior is that the bound taxanes do not establish any long-lasting hydrogen-bonding interactions with the amino acids making up this loop (βL275-βL286) so as to stabilize it into an α-helix, as epothilone A and zampanolide do (*Prota et al., 2013a*). Therefore, the β-hairpin conformation of the βM loop may

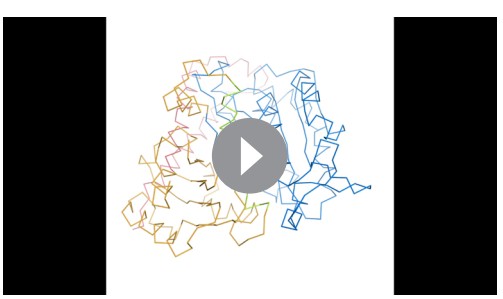

**Video 4.** Conformational transition from apo to 2a-bound, unassembled tubulin state. Luminal view on β-tubulin (view from the lumen in the context of a microtubule).

https://elifesciences.org/articles/84791/figures#video4

**Table 2.** Structural parameters of microtubules assembled in the presence of different nucleotides and drugs*.

| | Paclitaxel-pre microtubules | Paclitaxel-post microtubules | Paclitaxel-GDP tubulin | GDP-microtubules | GMPCPP-microtubules | Docetaxel-microtubules | Baccatin III-microtubules | 2a-microtubules | 2b-microtubules |
|---|---|---|---|---|---|---|---|---|---|
| Microtubule radius (nm) | 10.97±0.10 | 11.04±0.51 | 10.98±0.47 | 11.42±0.10 | 11.63±0.10 | 11.53±0.10 | 11.06±0.35 | 11.27±0.57 | 11.60±0.36 |
| Avg. PF number | 12.21±0.10 | 12.28±0.71 | 12.23±0.65 | 12.91±0.10 | 13.29±0.10 | 12.90±0.10 | 12.29±0.39 | 12.63±0.72 | 12.99±0.40 |
| Inter-PF distances (nm) | 5.58±0.01 | 5.59±0.33 | 5.57±0.29 | 5.50±0.03 | 5.45±0.03 | 5.57±0.01 | 5.61±0.18 | 5.55±0.31 | 5.56±0.17 |
| Avg. monomer length (nm) | 4.18±0.01 | 4.18±0.01 | 4.18±0.01 | 4.06±0.01 | 4.18±0.010 | 4.18±0.01 | 4.16±0.03 | 4.15±0.03 | 4.13±0.03 |
| 1 nm band peak position ($nm^{-1}$) | 6.02±0.01 | 6.02±0.01 | 6.02±0.01 | 6.19±0.01 | 6.02±0.01 | 6.02±0.01 | 6.04±0.5 | 6.06±0.05 | 6.08±0.05 |
| 8 nm band | Yes | Yes | Yes | No | Yes | Yes | Yes | Yes | Yes |

*Errors are SE of three independent.

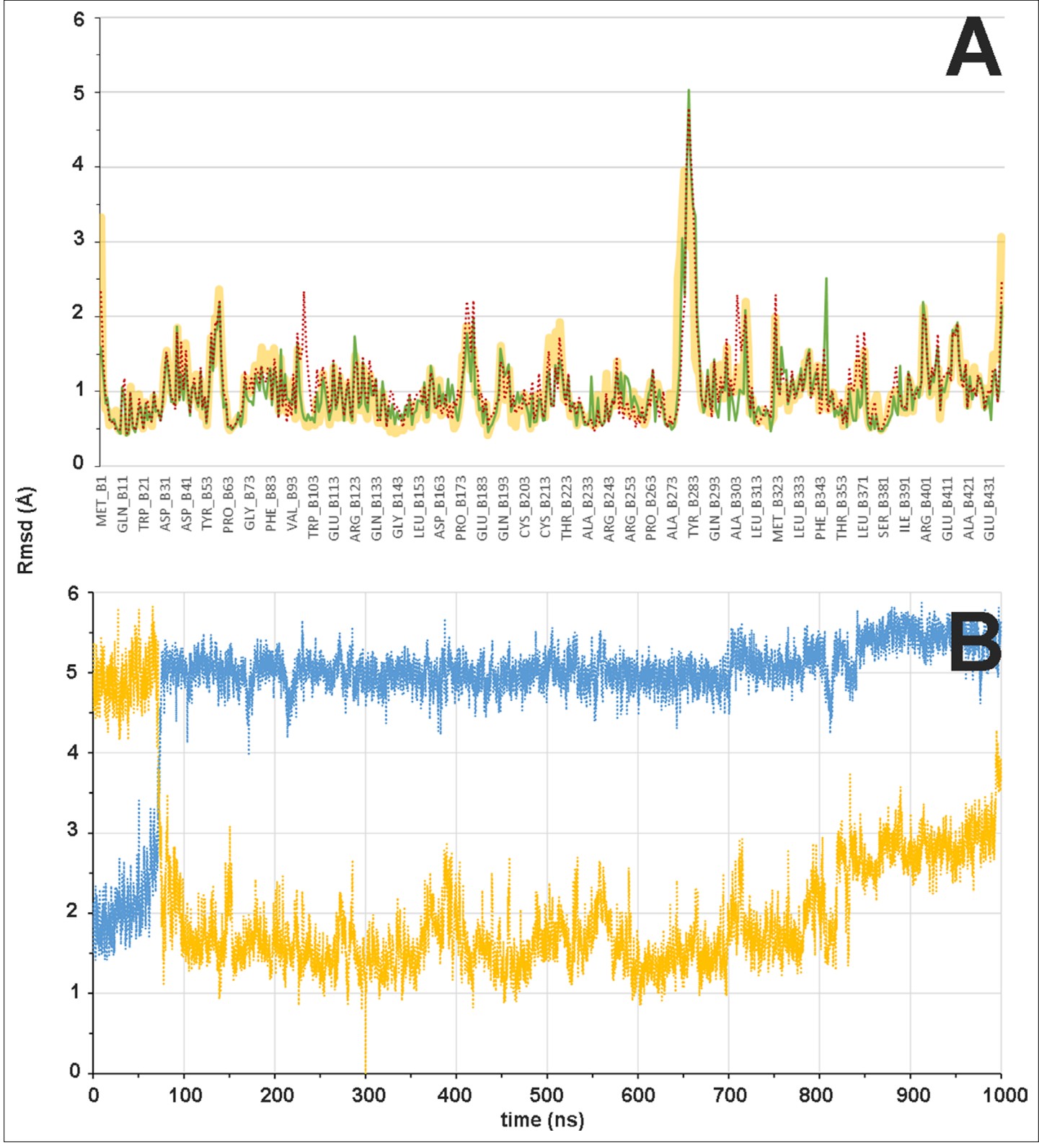

**Figure 8.** Flexibility of β subunit and βM loop during the αβ-tubulin dimer molecular dynamics (MD) simulation. (**A**) Mass-weighted positional fluctuations (or root-mean-square fluctuations, Å) by residue for atoms in the β subunit of the αβ-tubulin dimer over the course of 0.6 μs of MD simulation, in the apo form (yellow line) and in complex with baccatin III (green line) or paclitaxel (red dotted line). (**B**) Evolution of the conformation of the βM loop in the 1.0 μs simulation of the αβ-tubulin dimer free in solution. The Cα root-mean-square deviation is measured with respect to either the initial α-helical structure (blue line) or the extended hairpin conformation that was stabilized at 300 ns (orange line).

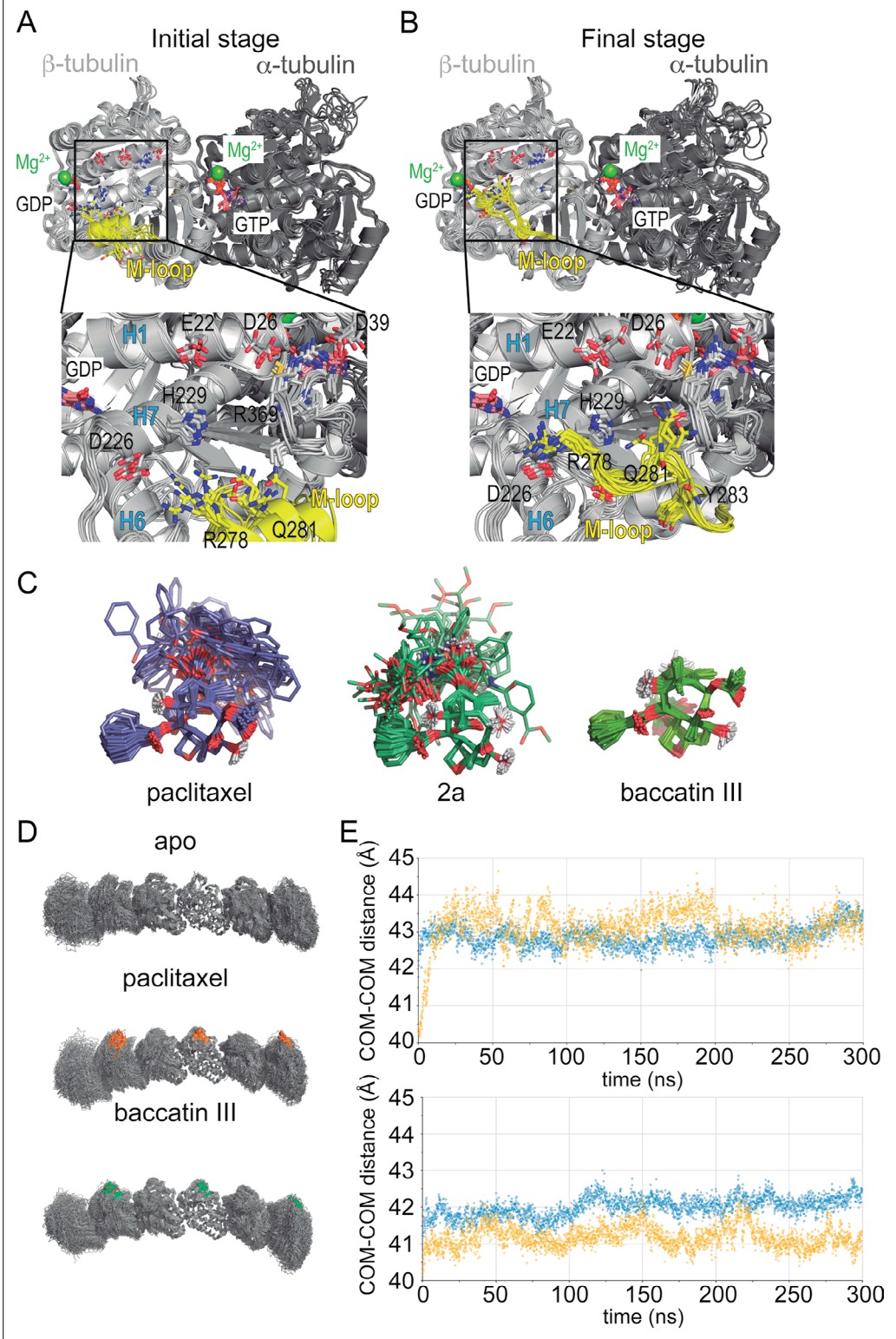

**Figure 9.** Molecular dynamics (MD) simulation of tubulin-taxane complexes. (**A,B**) MD simulations of the free αβ-tubulin dimer. (**A**) Initial stage of the simulation, starting from a βM loop (residues β275-β286; yellow) organized as an α-helix akin to what is observed in a microtubule and (**B**) after 100 ns of an MD simulation. (**C**) Overlaid snapshots taken every 5 ns over the course of a 250 ns MD simulation of paclitaxel (left), **2a** (middle), or baccatin III (right). (**D**) Snapshots of the protofilament model bound to paclitaxel or baccatin III and apo form. (**E**) Time evolution of the intermonomer distances (measured between the respective centers of mass; α2-β2 in blue and β2-α3 in yellow) in the simulated apo (lower graph) and liganded (upper graph) protofilaments.

compete efficiently with the binding of ligands to the taxane site (Animation 1). As a consequence, the free energy of ligand binding to tubulin dimers with a βM loop partially occluding the taxane site would be expected to be much lower (in the $10^3$–$10^4$ M$^{-1}$ range) than the free energy of binding to microtubules, as is indeed the case (*Díaz et al., 1993*; *Canales et al., 2014*; *Canales et al., 2011*). When considering paclitaxel, entry into the taxane site is further hampered by the fact that this bulky and highly hydrophobic molecule can adopt alternatively collapsed conformations in solution that are different from its bioactive, tubulin-bound T-shape conformation (*Snyder et al., 2001*; *Coderch et al., 2013*). The alternative paclitaxel conformations that are inexistent in the case of baccatin III or **2a** further reduce the apparent binding affinity below the solubility limit of the ligand (*Figure 9C*). These considerations might explain why we failed to obtain crystal structures of tubulin-paclitaxel and tubulin-docetaxel complexes. Conversely, we think that the less stringent requirements of the less bulky baccatin III and **2a** molecules to bind as compared to paclitaxel (*Figure 10*) may explain the success in obtaining co-crystal structures with tubulin.

The intermolecular hydrogen bond involving the oxetane O5 and the backbone NH of βT276 is a common feature to all three tubulin-taxane complexes, both in crystals (baccatin III and **2a**) and throughout the simulated MD trajectories (all three ligands). Paclitaxel and **2a** establish two other long-lived hydrogen bonds during our simulations, namely O4':(NE2)βH229 and O2':(O=C)βR369, which may involve – depending on context – a βR369-βG370 backbone rearrangement. In turn, the hydroxyl group at C13 of baccatin III alternates between acting as a direct or water-mediated hydrogen bond donor or an acceptor to/from (NE2)βH229 and (O=C)βR369, respectively. In the case of **2a** (and **2b**), on the other hand, it seems that the smaller and more flexible substituents at the C3' position – relative to those present in paclitaxel and docetaxel – allow an adaptation of the βR369-βG370 backbone in the crystal lattice that does not appear to be feasible for the pharmacologically used taxanes.

It has been reported previously that paclitaxel binds better to straight tubulin and promote tubulin assembly (i.e., they lower the critical concentration for tubulin assembly *Díaz et al., 1993*; *Buey et al., 2005*) being therefore able to prevent the straight-to-curved conformational transition in GDP-bound microtubules (*Elie-Caille et al., 2007*). However, our simulations indicate that protofilaments are curved both in the absence and in the presence of paclitaxel (*Figure 9D*), which suggests little or no direct influence of taxanes on the straight-to-curved conformational transition of tubulin. On the other hand, and similar to unassembled tubulin, although in our simulations of the taxane-bound protofilament the occupancy of the taxane site by the ligand constrains the available conformational space of the βM loop compared to that of the apo form, the loop still fails to adopt a well-defined secondary structure in the absence of additional stabilizing interactions with a neighboring protofilament.

Finally, we used a minimalist model of a solvated microtubule lattice in which we could study and compare two taxane-binding sites (β$_1$ and β$_1$), namely, an interfacial one that is highly preorganized for the binding of taxanes due to the stabilization of the βM loop into an α-helix by lateral lattice contacts (site 1), and another one that is fully exposed to the solvent (site 2). We found that the solvent-exposed paclitaxel-bound βM loop is not permanently structured as an α-helix, as expected, and that the major ligand interactions at site 2 are essentially the same as in the paclitaxel-bound αβ-tubulin heterodimer and the protofilament model (*Figure 10*). On the other hand, in site 1 dispersion forces, additional H-bonds, the hydrophobic effect, and decreased ligand entropy confer to the studied compounds (paclitaxel, baccatin III, and **2a**) higher binding free energies and longer residence times (i.e., lower $k_{off}$ values) relative to the tubulin dimer and the exposed taxane site 2 (*Figure 11*). The three H-bond-mediated anchoring points, namely O5:(NH)βT276, O4':(NE2)βH229, and O2':(O=C)βR369, are the same as those observed in the microtubule-paclitaxel complex structure (*Kellogg et al., 2017*). The hydrogen bond between the amide carbonyl O4' and the imidazole N$^\varepsilon$ of βHis229 is maintained in all the **2a** and paclitaxel complexes studied even though this interaction fluctuates substantially, as does the stacking of βHis229 on the benzoyl phenyl ring. However, the most important interaction that is strengthened laterally when a taxane is bound is that involving βTyr283, whose position in the βM loop is fixed by segment [85]QIFR[88] of loop βT3 from the neighboring β-tubulin subunit, as seen previously for other taxane-site ligands like, for example, zampanolide and taccalonolide AJ (*Sánchez-Murcia et al., 2019*; *Figure 11*).

Importantly, our simulations consistently reproduce the axial lattice expansion observed upon paclitaxel binding (*Arnal and Wade, 1995*; *Alushin et al., 2014*; *Estévez-Gallego et al., 2020*; *Kellogg et al., 2017*; *Debs et al., 2020*; *Manka and Moores, 2018*). We found that the expansion mainly

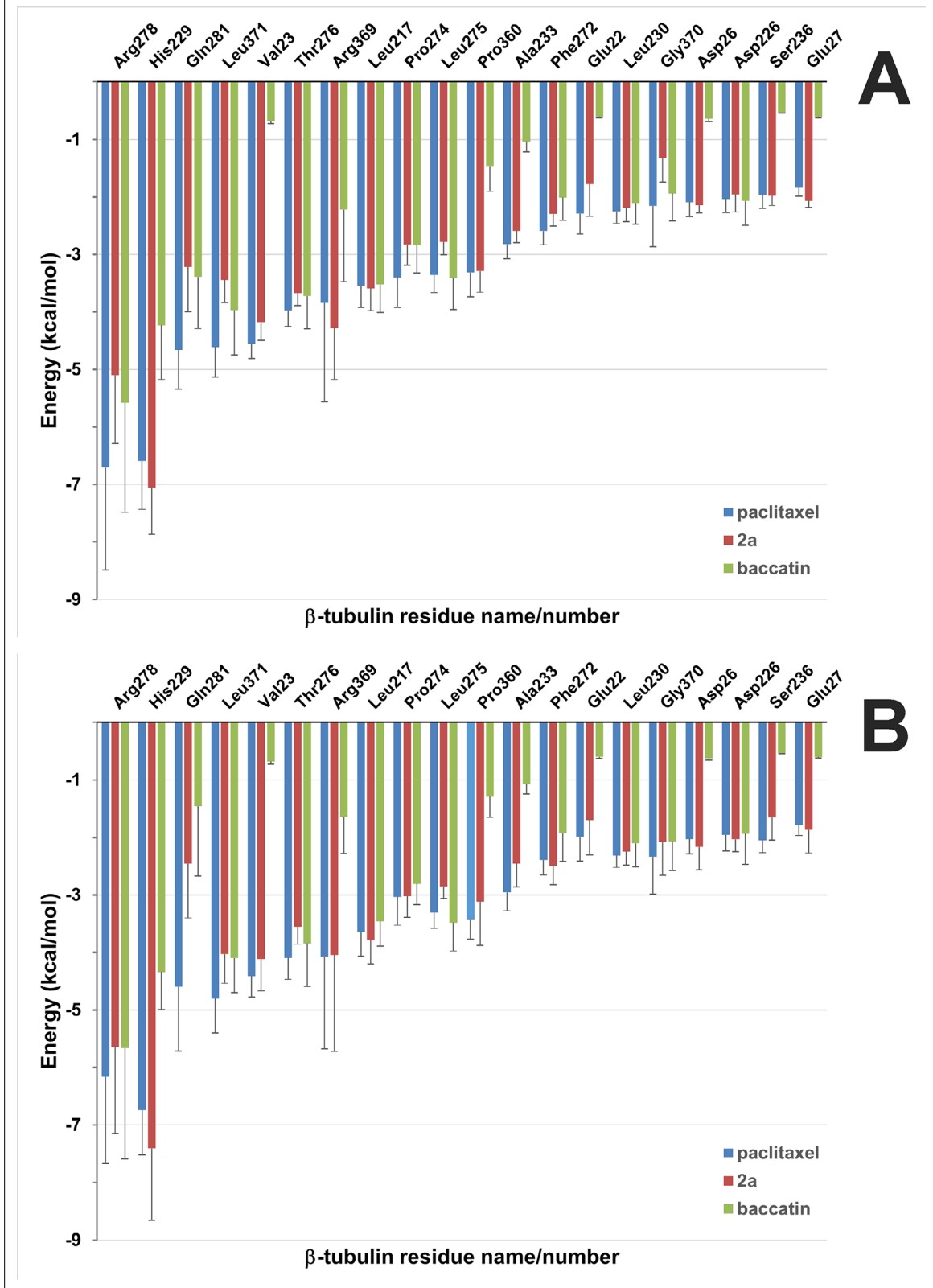

**Figure 10.** Solvent-corrected interaction energies between individual β2-tubulin residues and ligands throughout the molecular dynamics (MD) simulations of the minimalist representation of a microtubule. (**A**) The interfacial site 1 between neighboring protofilamentes. (**B**) The solvent-exposed site 2. These per-residue energies, which together represent a 'binding fingerprint', were calculated by means of the program MM-ISMSA (**Klett et al., 2012**) using 120 complex structures from the MD simulations after equilibration (5–600 ns), cooling down to 273 K and energy minimization. A cutoff of 1.5 kcal mol$^{-1}$ was used in the plot for enhanced clarity. Bars are standard errors.

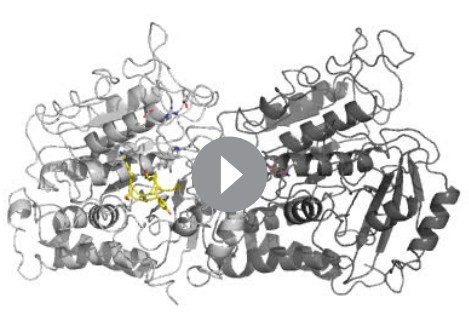

**Animation 1.** Molecular dynamics (MD) movie of the apo tubulin dimer showing the occupancy of the taxane site by the βM loop. 1 µs simulation, 1 snapshot every 5 ns, βM loop in yellow.

originates from displacement of the βS9-βS10 loop caused by the $\Phi/\Psi$ backbone rearrangement in the βR369-βG370 stretch. Because the βS9-βS10 loop acts as a lid covering and stapling the bound taxanes in their final location, this motion propagates toward the attached α-tubulin subunit so that the distance between the $\alpha_1$- and $\beta_2$-tubulin subunits of two longitudinally aligned, consecutive tubulin dimers increases by roughly 2 Å as compared to the unbound filament (*Figure 9E*), a feature that could not be detected in the taxane-bound crystal structures of curved tubulin.

Taken together, these analyses suggest that (i) taxanes bind better to the microtubule-assembled over the unassembled state of tubulin due to the preorganization of the βM loop that otherwise is stabilized in conformations that are incompatible with high-affinity taxane binding; (ii) the bulky C13 side chains preferentially recognize the assembled over the unassembled state of tubulin; (iii) the occupancy of the taxane site has no influence on the straightness of tubulin protofilaments; and (iv) the displacement of the βS9-βS10 loop of β-tubulin by the bound taxane results in microtubule expansion.

## Discussion

Previous studies on taxanes left us with several important open issues related to their molecular mechanism of microtubule stabilization. Why do they preferentially bind to the microtubule-assembled over the unassembled tubulin state? Are they involved in structuring of the βM loop, a molecular process that is required for microtubule assembly? Why do they distort/affect the microtubule lattice? Here, we used a combination of ligand engineering, structural biology, and computational approaches to gain insight into these pending questions.

First, we used a rational synthetic approach to dissect which parts of the paclitaxel molecule are involved in particular aspects of tubulin recognition and microtubule stabilization. Our results reveal that the baccatin III core of paclitaxel is responsible for filling most of the taxane site and for the key O5:NH(βT276) hydrogen-bonding interaction that is established between taxanes and β-tubulin. However, this interaction has only a marginal effect on the drug's microtubule-stabilizing effect (*Parness et al., 1982*; *Lataste et al., 1984*; *Kingston, 2000*), which requires the C13 side chain to increase the selectivity of the drugs for microtubules over unassembled tubulin. On the other hand, we found that ring A of paclitaxel precludes binding of the drug to the $T_2R$-TTL and TD1 crystals, while two taxanes with a modified, smaller C13 side chain (**2a** and **2b**) can bind due to the reduced size of their 3'-acylamino substituents and increased flexibility relative to paclitaxel. These smaller substituents allow the newly synthesized taxane derivatives to bind to curved, unassembled tubulin – while keeping their binding poses very similar to that described for paclitaxel when bound to straight tubulin in microtubules – by allowing adaptation of the whole ligand to a rearranged βR369-βG370 backbone in the crystal lattice.

Regarding the selective recognition of microtubules by taxanes, we found that it arises from two different terms. The first one is the differential interaction of the bulky C13 side chains with straight and curved tubulin. Our structural analysis reveals that a major structural difference is the environment of the position occupied by the 3'-*N*-benzamido phenyl ring A moiety of paclitaxel in microtubules: in the $T_2R$-TTL-**2a** structure, the βR369 side chain occupies the same space as does the βD26 side chain in the context of the assembled tubulin conformation in microtubules. The C13 side chain is involved in the interaction with helix βH1 that is flanked by the βH1-βS2. Upon transition to the microtubule-assembled, straight tubulin state, this space is narrowed down by the side chains of βD26, βK19, βE22, and βH229 to form a favorable environment for the interaction with ring A, which may lock

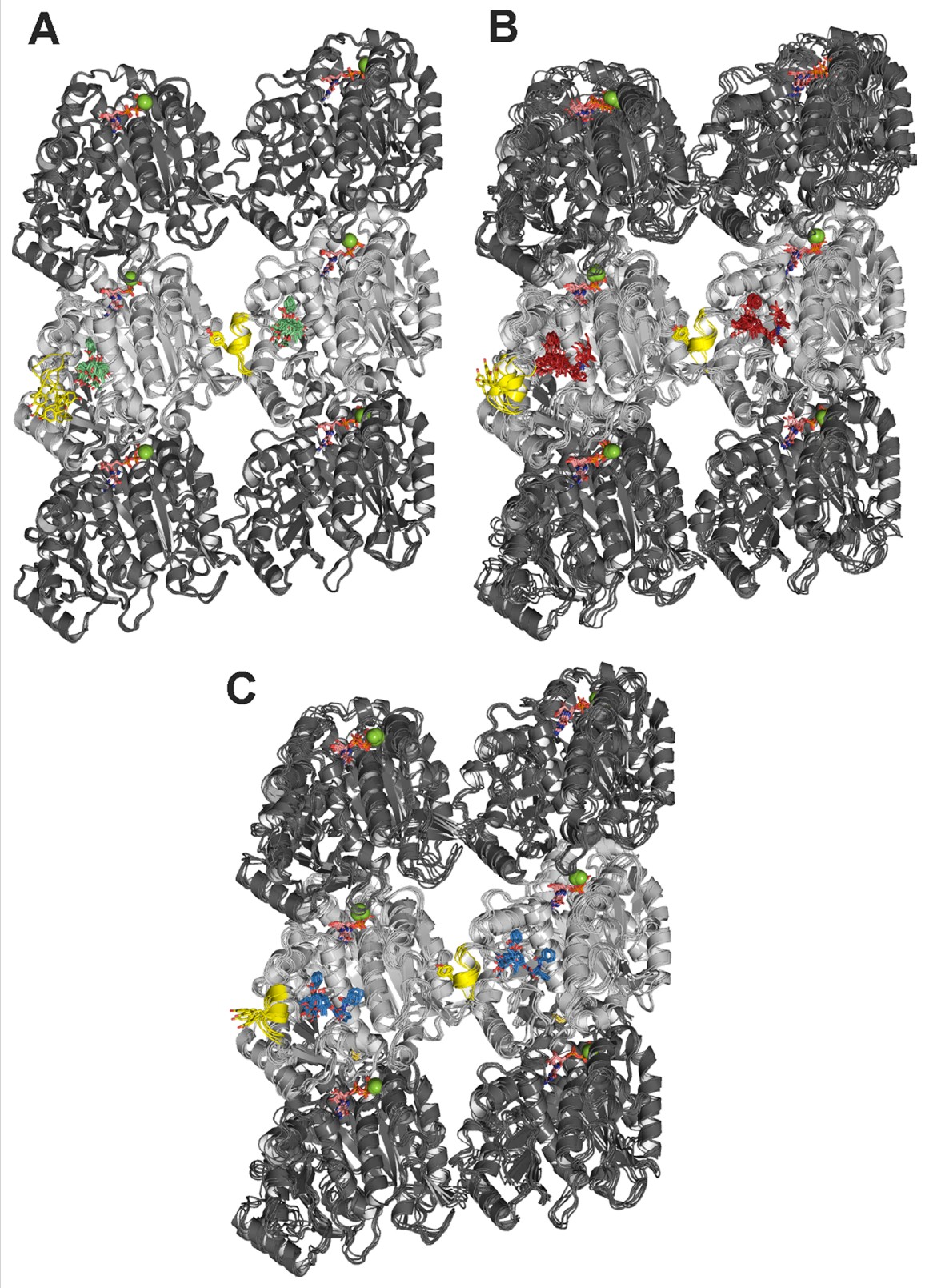

**Figure 11.** Molecular dynamics (MD simulations of minimalist) representations of a microtubule (($\alpha_1$-$\beta_1$-$\alpha_2$)/($\alpha_{1'}$-$\beta_{1'}$-$\alpha_{2'}$)) in complex with the ligands employed. Baccatin III (green, **A**), 2a (red, **B**), paclitaxel (blue, **C**). α- (dark gray) and β-tubulin (light gray) are displayed as ribbons, with the βM loop colored in yellow and the side chain of Tyr283 as sticks. Guanosine diphosphate (GDP) and guanosine triphosphate (GTP) are shown as sticks, with C atoms colored in salmon. $Mg^{2+}$ ions are displayed as green spheres. Each set of five overlaid structures represents a conformational ensemble made up

*Figure 11 continued on next page*

*Figure 11 continued*

of snapshots spaced by 5 ns taken from the equilibrated part of the trajectory and then cooled down to 273 K and energy minimized. Site 1 (at the top of each figure) is located at the interface between two neighboring protofilaments whereas site 2 (at the bottom of each figure) is devoid of any lateral contacts but exposed to the bulk solvent instead.

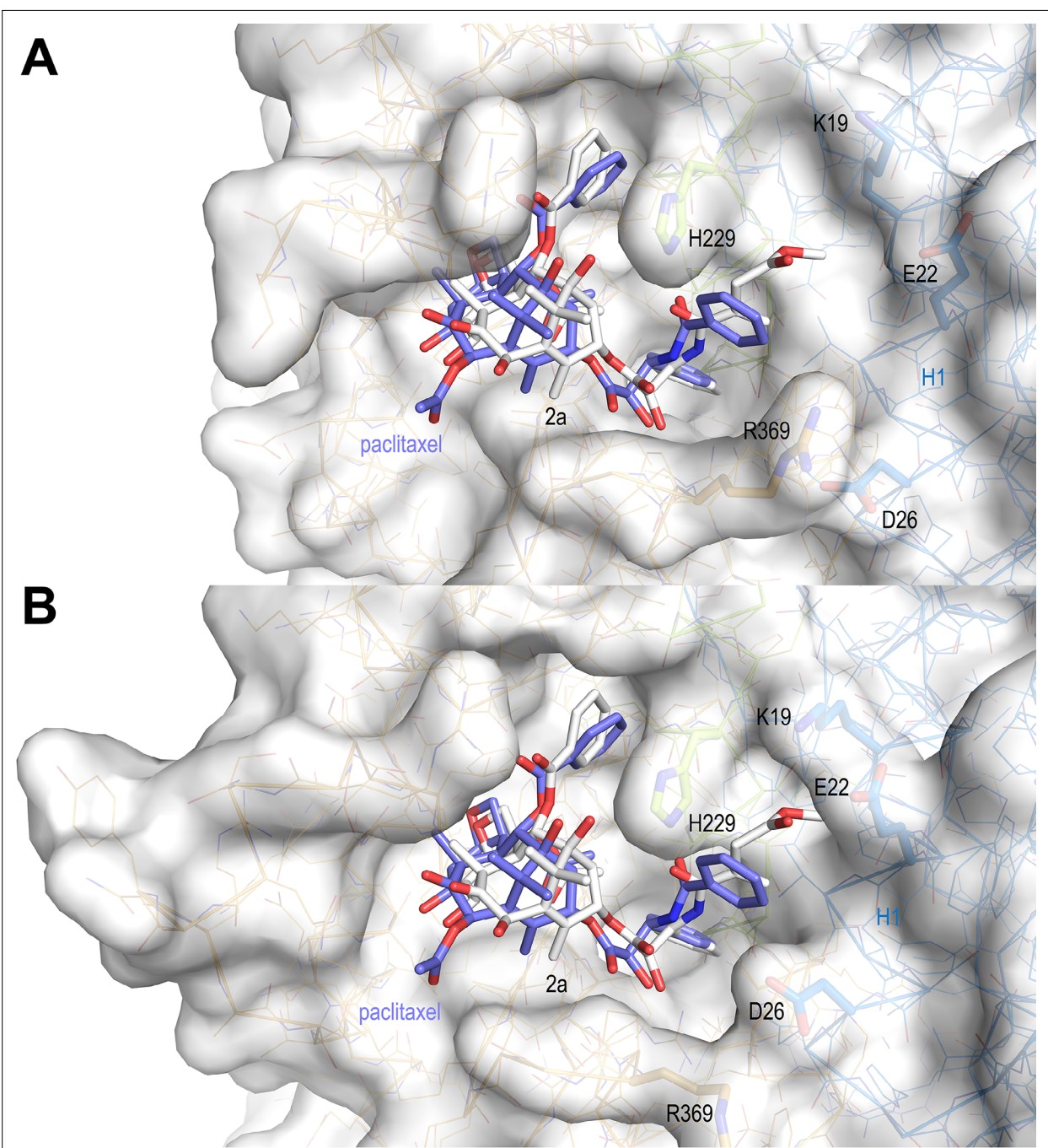

**Figure 12.** Surface representations of liganded taxane sites in both the curved and straight tubulin conformational states. (**A**) Curved tubulin; (**B**) straight tubulin. The structures of **2a** (white) and paclitaxel (slate) bound to microtubules (PDB ID 6WVR) were superimposed onto their central helices βH7. The side chains of the βM loop residue βR278 and of residues surrounding the C13 side chains of the ligands are in stick representation and are labeled. Helix βH1 is highlighted in ribbon representation.

the paclitaxel-bound tubulin in the straight conformation (*Figure 12B*). In the absence of the C13 side chain (baccatin III) or in the presence of less bulky and more 'articulated' moieties at the ring A position (**2a** and **2b**; their substituents at the N3 position have a rotatable bond in the middle), ligand binding is likely to be less affected by the curved-to-straight conformational transition, since much looser interactions can still be established with the charged residue side chains of the βS9-βS10 loop and helix H1 through water molecules (*Figure 3B*, *Figure 12A*). A second term that accounts for selectivity is occupancy of the taxane site by the βM loop in the absence of lateral contacts, which would be a general mechanism that accounts for the loss of at least four orders of magnitude in affinity when binding to unassembled tubulin relative to microtubules for all taxane-site ligands, including paclitaxel, docetaxel (*Díaz et al., 1993*), discodermolide (*Canales et al., 2011*), epothilones (*Canales et al., 2014*), and **2a** (this paper). Our MD simulations of the drug-free tubulin dimer shed new light on why taxanes and other taxane-site ligands bind tubulin dimers with affinities much lower than those reported for microtubules. Whereas in assembled microtubules the βM loop is structured as an α-helix and the preorganized taxane site is empty and ready to accommodate a ligand, in the unassembled tubulin dimer – as well as in the models of isolated protofilaments and the solvent-exposed site of the minimalist microtubule representation – this same loop displays a large conformational heterogeneity and can adopt a hairpin conformation that allows it to interact with the taxane site and thus to inhibit ligand binding (*Figure 11*). Moreover, when the tubulin dimer with the βM loop in an α-helical conformation was simulated in complex with baccatin III, **2a**, and paclitaxel, an evolution was systematically observed consisting of βM loop disordering similar to that likely responsible for the lack of electron density in the crystallographic apo structures. The MD analysis indicates that the conformational freedom of the βM loop in unassembled tubulin allows it to occupy the taxane-binding pocket in such a way as to preclude (or compete with) ligand binding. On the other hand, the free energy contribution of taxane-site ligands for microtubule assembly arises from the preferential recognition of the taxane-site conformation present in microtubules (*Nogales et al., 1999*).

Our results point to the βM loop as an essential structural element for the mode of action of paclitaxel and other clinically used taxanes. Our high-resolution structural analysis of baccatin III in complex with tubulin suggests that even this simplified taxane is able to reduce the flexibility of the βM loop by inducing a partial structuring of its N-terminal region. Further changes occur in the presence of a small C13 side chain, as in **2a** and **2b**, compared to paclitaxel, such as tilting the position of their baccatin III core region by ~20° within the binding pocket and inducing a subtle reorientation of tubulin domains with respect to one another. Despite the fact that we did not observe a complete structuring of the βM loop upon baccatin III, **2a**, or **2b** binding in their respective crystal structures or during MD simulations of free dimers and protofilaments, conformational changes were detected in this β-tubulin region that are in consonance with those observed upon paclitaxel binding to microtubules. Furthermore, our X-ray fiber diffraction studies indicate differences in interprotofilament contacts of shear-flow aligned microtubules bound to baccatin III, **2a**, or **2b**. This observation suggests that paclitaxel and the novel taxanes reported here indeed affect interprotofilament contacts so as to promote microtubule stability through interactions with the N-terminal section of the βM loop, in good agreement with observations reported previously (*Debs et al., 2020*; *Manka and Moores, 2018*).

Finally, we found that binding of taxanes to assembled microtubules results in a displacement of the βS9-βS10 loop, which promotes a lattice expansion. The description of the effect of paclitaxel on microtubule lattice parameters has been controversial. Initial analyses suggested that paclitaxel induces lattice expansion (*Arnal and Wade, 1995*; *Alushin et al., 2014*); however, subsequent studies reported only a minor effect (*Kellogg et al., 2017*; *Debs et al., 2020*; *Manka and Moores, 2018*). Our present results reinforce the view that lattice expansion is indeed a general consequence of taxane binding and does not require the presence of a C13 side chain. Since baccatin III is essentially biologically inert (*Parness et al., 1982*; *Lataste et al., 1984*; *Kingston, 2000*; *Andreu and Barasoain, 2001*), our data further indicate that lattice expansion is not an important factor contributing to the mechanism of microtubule stabilization by paclitaxel. Our MD analyses offer a plausible explanation for the taxane-induced longitudinal expansion of microtubules. Although in the complexes with **2a** and **2b** – but not in that with baccatin III – the crystal structures show that (NH)βG370 hydrogen bonds to the taxane side chain, the simulated complexes indicate that in solution it is the (O=C) βR369 that consistently acts as the hydrogen bond acceptor for the O2' hydroxyl of taxanes. In our view, these findings point to the βS9-βS10 loop as a major structural element that changes on taxane

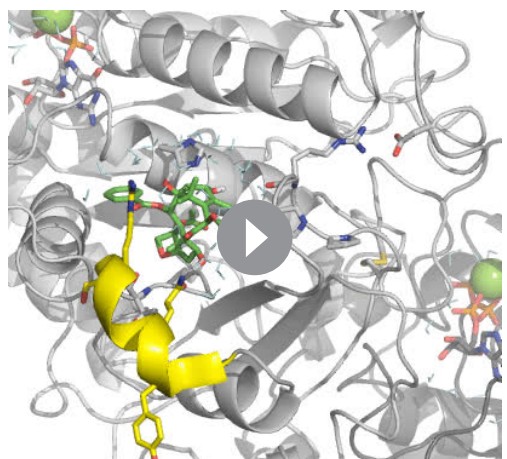

**Animation 2.** Simulation of ligand exit and entry using targeted molecular dynamics (MD) for baccatin III unbinding from and binding to αβ-tubulin.

binding, and this change is transmitted to the following α-tubulin subunits on both sides, hence the stretching or longitudinal expansion of the concatenated tubulin dimers. The fact that we observed the Φ/Ψ backbone rearrangement in the βR369-βG370 stretch upon cooling down the tubulin-paclitaxel and baccatin III complex obtained after the TMD procedure (Animation 2 and Animation 3) points to expansion of the cavity and consolidation of the hydrogen-bonding network as the main factors responsible for this conformational change.

In conclusion, our combined experimental and computational approach allowed us to describe the tubulin-taxane interaction in atomic detail and assess the structural determinants for binding. Our structural analyses further suggest a mode of action of paclitaxel by means of which its core moiety provides the main tubulin-interaction network while its C13 side chain enables selective recognition of the prestructured βM loop of the microtubule-assembled tubulin state. Such differential recognition is expected to promote microtubule formation and stabilization. On the other hand, the longitudinal expansion of the microtubule lattices arises from the accommodation of the taxane core within the site, a process that is, however, not related to the microtubule stabilization mechanism of taxanes.

## Materials and methods
### Proteins and ligands
Purified calf brain tubulin containing a mixture of isotypes (*Ludueña and Banerjee, 2008*) was obtained as described (*Andreu, 2007*) and used for biochemical, crystallographic, and fiber diffraction experiments. Paclitaxel (Taxol) was from Alfa Aesar Chemical, docetaxel (Taxotere) was kindly provided by Rhône Poulenc Rorer, Aventis (Schiltigheim, France), baccatin III was from Sigma, Flutax-2, Chitax 40, 3′-N-aminopaclitaxel (N-AB-PT) and Chitax 68 were synthesized as described (*Li et al., 2000*; *Matesanz et al., 2008*; *Ma et al., 2018*; *Díaz et al., 2000*). All compounds were diluted in 99.8% DMSO-D6 (Merck) to a final concentration of 20 mM and stored at –20°C. Their solubility in aqueous media was determined as described in *Sáez-Calvo et al., 2017*, Flutax-2 was found soluble, while a 100 µM solubility was found for docetaxel and a 50 µM for both paclitaxel and Chitax 40.

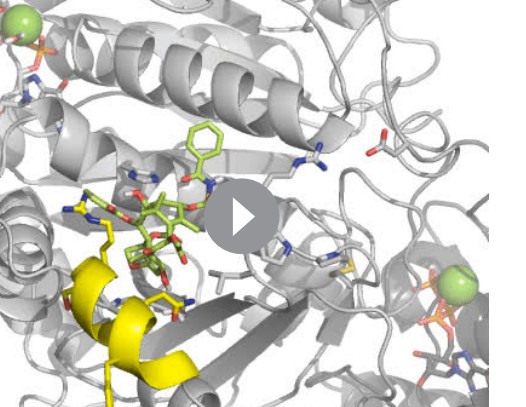

**Animation 3.** Simulation of ligand exit and entry using targeted molecular dynamics (MD) for paclitaxel unbinding from and binding to αβ-tubulin.

### Synthesis of taxoids 2a-2d (*Scheme 1*)

#### General experimental procedures
[1]H and [13]C NMR spectra were recorded on Varian 400, 500 MHz spectrometers, or a Bruker AVANCE III 600 MHz NMR spectrometer. Mass spectra (ESI) was measured on JEOL Accu TOF CS (JMS T100CS). Reagents were purchased from J&K and Alfa Aesar Chemical companies. All anhydrous solvents were purified and dried according to standard procedures, unless otherwise indicated. Reactions were monitored by TLC

**Scheme 1.** Reagents and conditions. (a) TESCl, triethylamine (TEA), 4-dimethylaminopyridine (DMAP), LiBr, THF, RT to 70°C, 84%; (b) LHMDS, THF, −45°C, 79%; (c) 10% Pd/C, $H_2$, MeOH, 64%; (d) DCC, DMAP, acid, DCM, 0°C to RT; (e) HF, Py, $CH_3CN$ or 5% HCl/MeOH, 32% for 2a, 40% for 2b, 48% for 2c, and 83% for 2e for two steps (d, e); (f) $PPh_3$, $CS_2$, THF, 83%.

(silica gel, GF254) with UV light and $H_2SO_4$-anisaldehyde spray visualization. The purity of the final compounds was analyzed by HPLC.

## 7,10-O-di(triethylsilyl)-10-deacetylbaccatin III (4)

To a stirred solution of **3** (1.82 g, 3.3 mmol) in anhydrous tetrahydrofuran (THF) (36 mL) under argon, 4-dimethylaminopyridine (DMAP) (400 mg, 3.3 mmol), triethylamine (8.3 mL, 69.4 mmol), and (chloro-triethylsilane) TESCl (4.5 mL, 26.4 mmol) were added dropwise. After the reaction mixture was stirred at room temperature (RT) for 5.5 hr, the solution of anhydrous LiBr (291 mg, 3.3 mmol) in anhydrous THF (1.8 mL) was added, the reaction mixture was refluxed at 65–70°C for 7 hr. Once cooled down, the mixture was diluted with ethyl acetate (200 mL). The mixture was washed with saturated aqueous $NaHCO_3$ solution (200 mL) and brine (200 mL), and dried over anhydrous $Na_2SO_4$. The organic layer was evaporated under reduced pressure. Purification of the crude product by silica gel chromatography (acetone:petroleum ether = 1:7) gave 84% yield of product **4** (2.13 g) as a light yellow oil: [1]H-NMR (400 MHz, CDCl$_3$): $\delta$ 0.55–0.71 p.p.m. (m, 12H), 0.94–1.02 (m, 18H), 1.04 (s, 3H), 1.18 (s, 3H), 1.65 (s, 3H), 1.85–1.91 (m, 1H), 2.01 (s, 3H), 2.22–2.28 (m, 5H), 2.49–2.57 (m, 1H), 3.91 (d, J=6.8 Hz, 1H), 4.14 (d, J=8.0 Hz, 1H), 4.27 (d, J=8.4 Hz, 1H), 4.42 (dd, J=10.4 Hz, 6.8 Hz, 1H), 4.81 (t, J=8.0 Hz, 1H), 4.93 (d, J=8.0 Hz, 1H), 5.21 (s, 1H), 5.61 (d, J=7.2 Hz, 1H), 7.45 (t, J=7.6 Hz, 2H), 7.57 (t, J=7.2 Hz, 1H), 8.09 (d, J=7.2 Hz, 2H). The [1]H NMR data are identical to those for 7,10-O-di(triethylsilyl)-10-deacetyl baccatin III in *Kung, 2012*.

## 7,10-O-di(triethylsilyl)-2′-O-(tert-butyldimethylsilyl)-3′-N-(de-tert-butoxycarbonyl)-3′-N-(benzyloxycarbonyl)docetaxel (6)

A stirred solution of **4** (2.12 g, 2.74 mmol) in anhydrous THF (35.7 mL) under argon was cooled to −45°C and lithium bis(trimethylsilyl)amide (LHMDS) (0.9 M in methylcyclohexane, 4.6 mL, 4.11 mmol) was added dropwise. The reaction mixture was stirred for 20 min at −45°C and then, the solution of **5** (*Ojima et al., 1995*) (1.352 g, 3.288 mmol) in anhydrous THF (9 mL) was added and the reaction mixture was stirred for 100 min at the same temperature. Afterward, the mixture was quenched with

saturated aqueous NH$_4$Cl solution (10 mL) and extracted with ethyl acetate (200 mL*2). The organic layer was washed with saturated aqueous NH$_4$Cl solution (100 mL) and brine (100 mL), dried over anhydrous Na$_2$SO$_4$. Solvent was removed under reduced pressure. Purification of the crude product by silica gel chromatography (acetone:petroleum ether = 1:10~1:7) gave 79% yield of product **6** (2.57 g) as a light yellow oil: $^1$H-NMR (400 MHz, CDCl$_3$): $\delta$ –0.31 p.p.m. (s, 3H), –0.08 (s, 3H), 0.56–0.72 (m, 12H), 0.75 (s, 9H), 0.95–1.03 (m, 18H), 1.20 (s, 3H), 1.22 (s, 3H), 1.69 (s, 3H), 1.84 (s, 3H), 1.88–1.96 (m, 2H), 2.33–2.39 (m, 1H), 2.49–2.54 (m, 4H), 3.86 (d, J=6.8 Hz, 1H), 4.21, 4.29 (ABq, J=8.4 Hz, each 1H), 4.41 (dd, J=10.4 Hz, 6.4 Hz, 1H), 4.55 (s, 1H), 4.93 (d, J=8.4 Hz, 1H), 4.97, 5.02 (ABq, J=12.4 Hz, each 1H), 5.16 (s, 1H), 5.37 (d, J=8.8 Hz, 1H), 5.67 (d, J=7.2 Hz, 1H), 5.72 (d, J=9.6 Hz, 1H), 6.25 (t, J=8.4 Hz, 1H), 7.20–7.32 (m, 8H), 7.38 (t, J=7.2 Hz, 2H), 7.48 (t, J=7.6 Hz, 2H), 7.57 (t, J=7.6 Hz, 1H), 8.13 (d, J=7.2 Hz, 2H); $^{13}$C-NMR (150 MHz, CDCl$_3$): $\delta$ −6.0,−5.4, 5.3, 5.9, 6.9, 10.5, 13.7, 14.1, 14.2, 18.1, 20.9, 21.0, 22.7, 23.1, 25.4, 26.5, 29.3, 29.6, 29.7, 31.9, 35.6, 37.3, 43.2, 46.6, 57.2, 58.3, 60.3, 66.8, 71.4, 72.6, 75.2, 75.3, 75.5, 76.7, 78.9, 81.2, 84.0, 126.4, 127.7, 127.8, 128.0, 128.4, 128.6, 129.5, 130.2, 133.4, 134.2, 136.3, 137.7, 138.6, 155.7, 167.0, 170.1, 171.1, 171.2, 205.2; ESIMS *m/z* 1184.6 [M+H]$^+$.

### 7,10-*O*-di(triethylsilyl)-2'-*O*-(*tert*-butyldimethylsilyl)-3'-*N*-(de-*tert*-butoxycarbonyl)docetaxel (**7**)

To a stirred solution of **6** (2.54 g, 2.14 mmol) in methanol (50 mL), 10% Pd/C (250 mg) was added under H$_2$ and the reaction mixture was stirred at RT for 20 hr. The mixture was diluted with methanol (50 mL), filtered and washed with methanol. The organic layer was evaporated under reduced pressure. Purification of the crude product by silica gel chromatography (acetone:petroleum ether = 1:8) gave 64% yield of product **7** (1.44 g) as a colorless oil with 18% yield of **6** (0.46 g) recovery: $^1$H-NMR (500 MHz, DMSO-$d_6$): $\delta$ –0.05 p.p.m. (s, 3H), –0.04 (s, 3H), 0.52–0.62 (m, 12H), 0.84 (s, 9H), 0.90–0.95 (m, 18H), 1.05 (s, 6H), 1.52 (s, 3H), 1.65–1.70 (m, 4H), 1.79–1.84 (m, 1H), 2.02–2.07 (m, 2H), 2.33 (s, 3H), 3.68 (d, J=7.0 Hz, 1H), 4.02–4.05 (m, 2H), 4.14, 4.30 (ABq, J=6.0 Hz, each 1H), 4.32 (dd, J=10.5 Hz, 6.5 Hz, 1H), 4.60 (s, 1H), 4.93 (d, J=9.5 Hz, 1H), 5.06 (s, 1H), 5.44 (d, J=7.0 Hz, 1H), 5.89 (t, J=9.0 Hz, 1H), 7.20–7.22 (m, 1H), 7.35–7.36 (m, 5H), 7.60 (t, J=7.5 Hz, 2H), 7.70 (t, J=7.5 Hz, 1H), 7.98 (d, J=7.0 Hz, 2H); $^{13}$C-NMR (125 MHz, DMSO-$d_6$): $\delta$ −5.4,−5.3, 4.8, 5.4, 6.7, 6.8, 10.1, 13.7, 17.9, 20.7, 22.6, 25.5, 26.3, 34.9, 36.8, 42.9, 45.9, 57.7, 58.9, 70.4, 72.4, 74.5, 75.0, 75.5, 76.6, 78.0, 80.0, 83.0, 124.2, 127.3, 128.0, 128.6, 129.5, 130.0, 133.4, 134.2, 137.1, 141.6, 165.2, 169.8, 172.1, 204.7; ESIMS *m/z* 1050.5 [M+H]$^+$.

### 3'-*N*-(de-*tert*-butoxycarbonyl)-3'-*N*-(4-methoxy-2-methylene-4-oxobutanoyl)docetaxel (**2a**)

To a stirred solution of **7** (43.6 mg, 0.042 mmol) in anhydrous dichloromethane (DCM) (0.34 mL) under argon, *N,N'*-dicyclohexylcarbodiimide (DCC) (17.1 mg, 0.083 mmol), DMAP (2.5 mg, 0.020 mmol), and the solution of itaconic acid monomethyl ester (*Ram and Meher, 2000*) (9.2 mg, 0.064 mmol) in DCM (0.15 mL) were added in ice bath. Then, the mixture was stirred for 2 hr at RT. The mixture was diluted with ethyl acetate (30 mL), filtered by celite and washed with ethyl acetate (30 mL). The organic layer was evaporated under reduced pressure. Purification of the crude product by silica gel chromatography (acetone:hexane = 1:9) gave crude product. Subsequently, to a stirred solution of the crude product in acetonitrile (1.7 mL), pyridine (1.0 mL, 12.1 mmol) and HF (0.52 mL, 12.1 mmol) was added and the reaction was stirred at RT for 24 hr. Following that, the mixture was diluted with ethyl acetate (50 mL), washed with brine (20 mL), extracted with ethyl acetate (20 mL), and dried over anhydrous Na$_2$SO$_4$. The organic layer was evaporated under reduced pressure. Purification of the crude product by silica gel chromatography (acetone:petroleum ether = 1:2) gave 32% yield (for two steps) of compound **2a** (11.0 mg) as a white solid: $^1$H-NMR (600 MHz, CD$_3$COCD$_3$): $\delta$ 1.11 p.p.m. (s, 3H), 1.20 (s, 3H), 1.69 (s, 3H), 1.80–1.84 (m, 1H), 1.88 (d, J=1.2 Hz, 3H), 2.16–2.20 (m, 1H), 2.30–2.34 (m, 1H), 2.39–2.45 (m, 4H), 3.34 (s, 2H), 3.61 (s, 3H), 3.89 (d, J=6.6 Hz, 1H), 4.13, 4.18 (ABq, J=8.4 Hz, each 1H), 4.25 (dd, J=11.4 Hz, 6.6 Hz, 1H), 4.69 (d, J=4.2 Hz, 1H), 4.95 (dd, J=9.6 Hz, 1.8 Hz, 1H), 5.24 (s, 1H), 5.53 (d, J=4.2 Hz, 1H), 5.65 (d, J=7.2 Hz, 1H), 5.73 (d, J=1.2 Hz, 1H), 6.15–6.18 (m, 2H), 7.27 (t, J=7.2 Hz, 1H), 7.38 (t, J=7.8 Hz, 2H), 7.46 (d, J=7.2 Hz, 2H), 7.54 (t, J=8.4 Hz, 2H), 7.64 (t, J=7.2 Hz, 1H), 8.09 (dd, J=8.4 Hz, 1.2 Hz, 2H); $^{13}$C-NMR (150 MHz, CD$_3$COCD$_3$): $\delta$ 9.1, 13.1, 20.2, 21.7, 25.9, 35.4, 36.1, 38.4, 42.9, 46.1, 51.1, 54.8, 57.2, 70.9, 71.0, 73.2, 73.8, 74.7, 75.6, 77.1, 80.5, 83.9, 126.8,

127.0, 127.5, 127.9, 128.2, 129.6, 129.9, 132.8, 134.6, 136.1, 137.3, 138.9, 165.5, 166.4, 169.3, 170.0, 172.4, 210.1; HRMS ($m/z$): [M+Na]$^+$ calcd for $C_{44}H_{51}NaNO_{15}$, 856.3259; found, 856.3157.

### 3'-N-(de-tert-butoxycarbonyl)-3'-N-(2-bromoacetyl)docetaxel (2b)

To a stirred solution of **7** (90 mg, 0.086 mmol) in anhydrous DCM (0.9 mL) under argon, DCC (53.2 mg, 0.26 mmol), DMAP (10.5 mg, 0.086 mmol), and the solution of bromoacetic acid (35.9 mg, 0.26 mmol) in DCM (0.1 mL) were added in ice bath. Then, the mixture was stirred for 2 hr at RT. The mixture was diluted with ethyl acetate (30 mL), filtered by celite, and washed with ethyl acetate (30 mL). The organic layer was evaporated under reduced pressure. Purification of the crude product by silica gel chromatography (acetone:hexane = 1:9) gave crude product (71 mg). Then, a stirred solution of the crude product (54 mg) was solved in 5% HCl/methanol (0.41 mL) in ice bath, and the reaction was stirred in ice bath for 1 hr and at RT for 12 hr. Afterward, the mixture was diluted with ethyl acetate (50 mL), washed with brine (20 mL), extracted with ethyl acetate (20 mL), and dried over anhydrous $Na_2SO_4$. The organic layer was evaporated under reduced pressure. Purification of the crude product by silica gel chromatography (acetone:petroleum ether = 1:1.5) gave 40% yield (for two steps) of compound **2b** (28.5 mg) as a white solid: $^1$H-NMR (600 MHz, CD$_3$COCD$_3$): $\delta$ 1.11 p.p.m. (s, 3H), 1.18 (s, 3H), 1.69 (s, 3H), 1.80–1.84 (m, 1H), 1.88 (d, $J$=1.2 Hz, 3H), 2.16–2.20 (m, 1H), 2.29–2.33 (m, 1H), 2.38 (s, 3H), 2.40–2.45 (m, 1H), 3.89 (d, $J$=7.2 Hz, 1H), 3.95, 4.00 (ABq, $J$=12.0 Hz, each 1H), 4.13, 4.18 (ABq, $J$=8.4 Hz, each 1H), 4.23 (dd, $J$=11.4 Hz, 6.6 Hz, 1H), 4.70 (d, $J$=4.2 Hz, 1H), 4.95 (dd, $J$=9.6 Hz, 1.8 Hz, 1H), 5.23 (s, 1H), 5.49 (d, $J$=4.2 Hz, 1H), 5.65 (d, $J$=7.2 Hz, 1H), 6.16 (t, $J$=8.4 Hz, 1H), 7.28 (t, $J$=7.2 Hz, 1H), 7.39 (t, $J$=7.2 Hz, 2H), 7.47 (d, $J$=7.2 Hz, 2H), 7.56 (t, $J$=7.8 Hz, 2H), 7.65 (t, $J$=7.2 Hz, 1H), 8.09 (d, $J$=8.4 Hz, 2H); $^{13}$C-NMR (150 MHz, CD$_3$COCD$_3$): $\delta$ 9.1, 13.1, 20.2, 21.7, 25.8, 35.4, 36.0, 42.8, 46.0, 55.3, 57.2, 59.4, 70.9, 73.0, 73.7, 74.7, 75.6, 77.1, 80.5, 83.8, 126.8, 127.2, 128.0, 128.1, 129.5, 129.9, 132.8, 136.1, 137.2, 138.3, 165.4, 165.9, 170.0, 172.1, 210.0; HRMS ($m/z$): [M+Na]$^+$ calcd for $C_{40}H_{46}NaBrNO_{13}$, 850.2153; found, 850.2037.

### 3'-N-(de-tert-butoxycarbonyl)-3'-N-(2-iodoacetyl)docetaxel (2c)

Taxoid **2c** was synthesized with iodoacetic acid following the similar procedure for **2b**. Yield of 48% (for two steps), 28.5 mg, white solid: $^1$H-NMR (600 MHz, CD$_3$COCD$_3$): $\delta$ 1.12 p.p.m. (s, 3H), 1.19 (s, 3H), 1.71 (s, 3H), 1.80–1.85 (m, 1H), 1.90 (d, $J$=1.2 Hz, 3H), 2.22–2.26 (m, 1H), 2.34–2.38 (m, 1H), 2.40 (s, 3H), 2.41–2.46 (m, 1H), 3.82, 3.87 (ABq, $J$=9.6 Hz, each 1H), 3.91 (d, $J$=7.2 Hz, 1H), 4.14, 4.19 (ABq, $J$=8.4 Hz, each 1H), 4.27 (dd, $J$=10.8 Hz, 6.6 Hz, 1H), 4.71 (d, $J$=3.6 Hz, 1H), 4.95 (dd, $J$=9.6 Hz, 2.4 Hz, 1H), 5.23 (s, 1H), 5.51 (d, $J$=3.6 Hz, 1H), 5.66 (d, $J$=7.2 Hz, 1H), 6.20 (t, $J$=9.0 Hz, 1H), 7.29 (t, $J$=7.2 Hz, 1H), 7.39 (t, $J$=7.2 Hz, 2H), 7.48 (d, $J$=7.2 Hz, 2H), 7.56 (t, $J$=7.8 Hz, 2H), 7.65 (t, $J$=7.8 Hz, 1H), 8.11 (d, $J$=9.0 Hz, 1.8 Hz, 2H); $^{13}$C-NMR (150 MHz, CD$_3$COCD$_3$): $\delta$ 9.3, 13.3, 20.4, 21.9, 26.1, 35.6, 36.3, 43.0, 46.3, 55.3, 57.4, 59.6, 71.1, 71.2, 73.2, 74.0, 74.9, 75.8, 77.4, 80.7, 84.0, 127.0, 127.3, 128.2, 128.3, 129.8, 130.1, 133.0, 136.3, 137.4, 138.7, 165.7, 167.7, 170.2, 172.4, 210.2; ESIMS $m/z$ 876.2 [M+H]$^+$, 898.2 [M+Na]$^+$.

### 3'-N-(de-tert-butoxycarbonyl)-3'-N-(2-azidoacetyl)docetaxel (2e)

Taxoid **2e** was synthesized with azidoacetic acid (*Brabez et al., 2011*) following the similar procedure for **2a**. Yield of 83% (for two steps), 25.0 mg, colorless oil: $^1$H-NMR (500 MHz, CD$_3$COCD$_3$): $\delta$ 1.12 p.p.m. (s, 3H), 1.18 (s, 3H), 1.69 (s, 3H), 1.79–1.85 (m, 1H), 1.87 (s, 3H), 2.12–2.17 (m, 1H), 2.27–2.32 (m, 1H), 2.36 (s, 3H), 2.40–2.46 (m, 1H), 3.89 (d, $J$=7.0 Hz, 1H), 4.01 (s, 2H), 4.13, 4.17 (ABq, $J$=8.0 Hz, each 1H), 4.25 (dd, $J$=11.0 Hz, 6.5 Hz, 1H), 4.67 (d, $J$=4.5 Hz, 1H), 4.95 (d, $J$=8.0 Hz, 1H), 5.23 (s, 1H), 5.50 (d, $J$=4.5 Hz, 1H), 5.65 (d, $J$=7.5 Hz, 1H), 6.16 (t, $J$=9.0 Hz, 1H), 7.28 (t, $J$=7.5 Hz, 1H), 7.39 (t, $J$=7.5 Hz, 2H), 7.47 (d, $J$=7.5 Hz, 2H), 7.56 (t, $J$=7.5 Hz, 2H), 7.65 (t, $J$=7.0 Hz, 1H), 8.09 (d, $J$=7.5 Hz, 2H); $^{13}$C-NMR (125 MHz, CD$_3$COCD$_3$): $\delta$ 9.3, 13.3, 20.4, 22.0, 26.0, 35.7, 36.3, 43.0, 46.3, 51.3, 55.4, 57.4, 71.0, 71.2, 73.5, 74.0, 74.9, 75.8, 77.4, 80.7, 84.1, 127.1, 127.5, 128.3, 128.4, 129.8, 130.1, 133.0, 136.4, 137.4, 138.8, 165.6, 167.2, 170.2, 172.4, 210.3; ESIMS $m/z$ 813.3 [M+Na]$^+$.

### 3'-N-(de-tert-butoxycarbonyl)-3'-N-(2-isothiocyanatoacetyl)docetaxel (2d)

To a stirred solution of **2e** (16.4 mg, 0.021 mmol) in anhydrous THF (0.32 mL) under argon, Ph$_3$P (8.5 mg, 0.032 mmol) and CS$_2$ (12.6 µL, 0.21 mmol) were added and the mixture was stirred for 50 hr

at RT. The mixture was evaporated under reduced pressure. Purification of the crude product by silica gel chromatography (acetone:hexane = 1:9) gave 83% yield of compound **2d** (28.5 mg) as a white solid: $^1$H-NMR (600 MHz, CD$_3$COCD$_3$): $\delta$ 1.08 p.p.m. (s, 3H), 1.10 (s, 3H), 1.67 (s, 3H), 1.76–1.84 (m, 5H), 2.06–2.09 (m, 1H), 2.36 (s, 3H), 2.40–2.45 (m, 1H), 3.85 (d, $J$=7.2 Hz, 1H), 4.11, 4.15 (ABq, $J$=7.8 Hz, each 1H), 4.18–4.25 (m, 3H), 4.94 (dd, $J$=9.6 Hz, 1.8 Hz, 1H), 5.20 (s, 1H), 5.59 (d, $J$=7.2 Hz, 1H), 5.65 (d, $J$=6.6 Hz, 1H), 6.02 (t, $J$=9.0 Hz, 1H), 6.17 (d, $J$=10.2 Hz, 1H), 7.25 (t, $J$=7.2 Hz, 1H), 7.38 (t, $J$=7.8 Hz, 2H), 7.54 (d, $J$=7.8 Hz, 2H), 7.61 (t, $J$=7.8 Hz, 2H), 7.69 (t, $J$=7.8 Hz, 1H), 8.04 (dd, $J$=8.4 Hz, 1.2 Hz, 2H); $^{13}$C-NMR (150 MHz, CD$_3$COCD$_3$): $\delta$ 9.0, 13.0, 19.9, 21.7, 25.6, 35.2, 35.9, 42.6, 46.0, 47.4, 57.1, 59.4, 69.7, 69.8, 70.8, 73.6, 74.5, 75.5, 77.0, 80.3, 83.8, 128.0, 128.1, 128.5, 129.4, 129.8, 132.9, 135.5, 136.1, 136.9, 165.3, 169.9, 171.7, 173.3, 209.8; HRMS (*m/z*): [M+Na]$^+$ calcd for C$_{41}$H$_{46}$NaN$_2$O$_{13}$S, 829.2721; found, 829.2619.

## Crystallization, data collection, and structure determination

Crystals of T$_2$R-TTL were generated as described (*Prota et al., 2013a*; *Prota et al., 2013b*). Suitable T$_2$R-TTL crystals were soaked for 8 hr in reservoir solutions (2–4% PEG 4 K, 2–10% glycerol, 30 mM MgCl$_2$, 30 mM CaCl$_2$, 0.1 M MES/imidazole pH 6.7) containing either 10 mM baccatin III, 5 mM **2a** or **2b**. Subsequently, crystals were flash cooled in liquid nitrogen following a brief transfer into cryo solutions containing the reservoir supplemented with 16% and 20% glycerol. All data were collected at beamline X06DA at the Swiss Light Source (Paul Scherrer Institut, Villigen PSI, Switzerland). Images were indexed and processed using XDS (*Kabsch, 2010*). Structure solution using the difference Fourier method and refinement were performed using PHENIX (*Adams et al., 2010*). Model building was carried out iteratively using the Coot software (*Emsley et al., 2010*). Data collection and refinement statistics for all three T$_2$R-TTL-complexes are given in *Table 1*. Molecular graphics and analyses were performed with PyMol (The PyMOL Molecular Graphics System, Version 2.3.2, Schrödinger, LLC). To compare the structures of both baccatin III and **2a** complexes in the curved tubulin conformation to the straight tubulin in paclitaxel stabilized microtubule (PDB ID 6WVR), all structures were superimposed onto the taxane site of **2a** (residues 208–219+225-237+318–320+359–376+272–276+287–296; rmsd$_{BacIII}$ 0.171 Å [48 C$_\alpha$ atoms], rmsd$_{5SYF}$ 0.868 Å [52 C$_\alpha$ atoms]).

## Biochemistry

The binding constants of both **2a** and baccatin III to unassembled dimeric tubulin were measured by centrifugation. Increasing amounts of dimeric tubulin (up to 150 µM) prepared in NaPi-GTP buffer (10 mM sodium phosphate, 0.1 mM GTP, pH 7.0) were incubated with a fixed concentration (50 µM) of either baccatin III or **2a**, incubated for 30 min at 25°C and centrifuged at 100,000 rpm in a TLA-100.2 rotor for 2 hr at 25°C. Then, samples were divided into upper (100 µL) and lower (100 µL) parts and 100 µL of NaPi were added to both of them. Afterward, 10 µM of either docetaxel or paclitaxel were added as internal standard and samples were subjected three times to an organic extraction using DCM (v:v). DCM was removed by evaporation and samples were resuspended in methanol 70%. Finally, ligand content was analyzed using an HPLC system (Agilent 1100 Series) and samples were separated using a Zorbax Eclipse XDB-C18 column (methanol 70% isocratic condition; 20 min runs). Tubulin content was determined by BCA for each sample. Ligand concentration in the upper 100 µL was considered as free concentration, while this in the lower 100 µL was considered as the sum of bound and free concentrations. Binding constants of tubulin for the ligand were calculated assuming a single binding site per tubulin dimer using SIGMAPLOT 14.5 Sigmastat Software Inc.

## Microtubule shear-flow alignment and X-ray fiber diffraction experiments

X-ray fiber diffraction data were collected in BL11-NDC-SWEET beamline of ALBA Synchrotron at a $\lambda$=0.827 nm as described in *Estévez-Gallego et al., 2020*. Radial structural parameters (microtubule diameter and average inter-PT distances) and dimer/monomer length (from the fourth harmonic of the first layer-line signals) were determined as described in *Estévez-Gallego et al., 2020*.

## Molecular modeling

### In silico model building and MD simulations

Our reduced representation of a straight microtubule for simulation purposes consisted of the $\alpha_1$:$\beta_1$:$\alpha_2$ subunits from one protofilament together with the closely interacting $\alpha_{1'}$:$\beta_{1'}$:$\alpha_{2'}$ subunits from a neighboring protofilament, as found in the cryo-EM reconstruction of an undecorated microtubule in complex with zampanolide (PDB ID 5SYG, 3.5 Å resolution, α1β2 isotype) (*Kellogg et al., 2017*). Likewise, $\alpha_1$:$\beta_1$ made up the isolated dimer, and three concatenated α:β dimers provided the starting straight protofilament. Missing residues 39–48 in the four α subunits were added, and the partially hydrated $Ca^{2+}$ coordinated by Asp39, Thr41, Gly44, and Glu55 was replaced by $Mg^{2+}$. Computation of the protonation state of titratable groups at pH 6.8 and addition of hydrogen atoms to each protein ensemble were carried out using the H++ 3.0 Web server (*Anandakrishnan et al., 2012*). Nonetheless, in agreement with previous work from our group, the side chain carboxylic group of βGlu200 in the colchicine-binding site was considered to be protonated (*Bueno et al., 2018*) and a disulfide bond was created to link the side chains of βCys241 and βCys356 (*Sánchez-Murcia et al., 2019*). The four GTP and two GDP molecules in the nucleotide-binding sites of α- and β-tubulin, respectively, were kept, together with their coordinated $Mg^{2+}$ ions and hydrating water molecules. For consistency with the Protein Data Bank, residue numbering and secondary structure assignment herein follow the α-tubulin-based definitions given by *Löwe et al., 2001*.

The initial molecular models of the taxane complexes were built by best-fit superposition of β-tubulin in their respective crystallographic complexes, as reported here for baccatin (PDB ID 8BDE), and **2a** (PDB ID 8BDF) – and previously for paclitaxel (PDB ID 1JFF)– (*Coderch et al., 2013*), onto the microtubule, protofilament, or α:β dimer structure described above. Ab initio geometry optimization of baccatin, paclitaxel, and **2a**, followed by derivation of atom-centered RESP charges (*Wang et al., 2000*) was achieved using a 6–31G* basis set, the density functional tight-binding method, and the IEF-SCRF continuum solvent model (*Scalmani and Frisch, 2010*) for water, as implemented in program Gaussian 09 (Revision D.01) (*Frisch, 2009*). The *gaff* (*Wang et al., 2004*) and *ff14SB* (*Maier et al., 2015*) AMBER force fields were used for ligand and protein atoms, respectively. The molecular graphics program PyMOL (Version 1.8, Schrödinger, LLC) was employed for structure visualization, molecular editing, and figure preparation.

All the ligand:tubulin complexes and their respective apo forms were solvated into a cubic box of TIP3P water molecules – with a minimal distance of the protein to the borders of 12 Å – and neutralized by addition of a sufficient number of $Na^+$ ions. These ensembles were simulated under periodic boundary conditions and electrostatic interactions were computed using the particle mesh Ewald method (*Salomon-Ferrer et al., 2013*) with a grid spacing of 1 Å. The cutoff distance for the non-bonded interactions was 9 Å and the SHAKE algorithm (*Ryckaert et al., 1977*) was applied to all bonds involving hydrogens so that an integration step of 2.0 fs could be used. All hydrogens and water molecules were first reoriented in the electric field of the solute and then all protein residues, ligands, counterions, and waters were relaxed by performing 5000 steps of steepest descent followed by 50,000 steps of conjugate gradient energy minimization. The resulting geometry-optimized coordinate sets were used as input for the MD simulations at a constant pressure of 1 atm and 300 K using the *pmemd.cuda_SPFP* engine (*Le Grand et al., 2013*) as implemented in AMBER 18 for GeForce Nvidia GTX 980 graphics processing units. Ligands, water molecules, and counterions were first relaxed around the positionally restrained protein (1 kcal mol$^{-1}$ Å$^{-2}$ on Cα atoms) during a heating and equilibration period lasting 0.5 ns. For the remaining simulation time (from 250 to 1200 ns depending on the system), the macromolecular ensembles were allowed to evolve and coordinates were collected every 0.1 ns for further analysis by means of the *cpptraj* module in AMBER (*Roe and Cheatham, 2013*). Positional restraints were used only in the case of the apo- and ligand-bound microtubule, in which case a weak harmonic restraint (0.5 mol$^{-1}$ Å$^{-2}$) on all Cα atoms (except for those in amino acids 276–374 of both β subunits that make up the βM loops and a large part of the α:β interfaces) was employed to preserve the overall architecture observed in the cryo-EM structure. Snapshots taken every 5 ns were cooled down from 300 to 273 K over a 1 ns period using a simulated annealing procedure (*Brunger and Adams, 2002*); the geometries of these 'frozen' complexes were then optimized by carrying out an energy minimization until the root-mean-square of the Cartesian elements of the gradient was less than 0.01 kcal mol$^{-1}$ Å$^{-1}$. The resulting ensembles of low-energy and geometrically optimized representative structures, which are expected to be closer to the global energy minima

(*Sánchez-Murcia et al., 2019*), were used to calculate the residue-based, solvent-corrected interaction energies.

## Geometry and binding energy analysis

Both the trajectory snapshots and the sets of representative optimized coordinates for each complex studied were analyzed in geometrical terms with the aid of the *cpptraj* routines (*Roe and Cheatham, 2013*) from the AmberTools18 suite. Estimations of the solvent-corrected binding energies were provided by our in-house MM-ISMSA software (*Klett et al., 2012*), which makes use of a sigmoidal, distance-dependent dielectric function (*Morreale et al., 2007*), and also provides a per-residue decomposition into van der Waals, coulombic, apolar, and desolvation contributions.

## Steered MD simulations

The macromolecular assemblies composed of an α:β dimer in complex with either baccatin or paclitaxel, as obtained after 5 ns of MD equilibration at 300 K, were additionally subjected to a targeted MD dynamics (tMD) procedure by means of which the trajectories were biased so as to force ligand exit first and then re-entry into the binding site. The tMD approach was followed essentially as described (*Rodríguez-Barrios et al., 2005*) and made use of the parallel implementation of the AMBER *sander. MPI* code running on four CPUs, which allows the solvent molecules to move freely and follow the dynamics of ligand and protein. A restraint was defined in terms of a mass-weighted root-mean-square (rms) superposition to the final reference structure (target) that is applied in the force field as an extra energy term of the form $E = 0.5 k_r N \left( rmsd - trmsd \right)^2$ , where $k_r$ is the force constant, $N$ is the number of atoms, and $trmsd$ is the target rms deviation. A negative value of the force constant (−0.5 kcal mol$^{-1}$ Å$^{-2}$ over 0.5 ns using only the ligand's oxygen atoms in the rms definition) was employed to force the ligand coordinates away from the initial docking location whereas a positive one was used to find a low-energy path leading from the unbound ligand obtained from the previous procedure back to the initial target structure.

Whereas the same value of 0.5 kcal mol$^{-1}$ Å$^{-2}$ over 0.5 ns proved sufficient to bring baccatin back to its binding pocket, it was considerably more cumbersome to achieve the same goal in the case of paclitaxel, in which case it was imperative to apply additional conformational restraints to fixate both the T-shape of the ligand and an α-helical βM loop for reasons discussed in the text.

## Acknowledgements

We thank Ganadería Fernando Díaz for calf brains supply and staff of beamlines X06DA of the Swiss Light Source (Paul Scherrer Institut, Villigen PSI, Switzerland) and BL11-NDC-SWEET (ALBA, Cerdanyola del Vallès, Spain) for their support. We also thank Mr Pedro Gascón Blanco for his private donation to the project to support a month of a student salary.

This article is dedicated to the memory of Dr Linda Amos, a dear friend and pioneer in the study of microtubules and the mechanism of action of paclitaxel (*Amos and Löwe, 1999*), who passed away while we were assembling this manuscript.

## Additional information

### Funding

| Funder | Grant reference number | Author |
| --- | --- | --- |
| Ministerio de Ciencia e Innovación | PID2019-104545RB-I00 (AEI/10.13039/501100011033) | J Fernando Díaz |
| Consejo Superior de Investigaciones Científicas | PIE 201920E111 | J Fernando Díaz |
| Fundación Tatiana Pérez de Guzmán el Bueno | Proyecto de Investigación en Neurociencia 2020 | J Fernando Díaz |

| Funder | Grant reference number | Author |
|--------|------------------------|--------|
| European Union NextGenerationEU | H2020-MSCA-ITN-2019 860070 TUBINTRAIN | Andrea E Prota J Fernando Díaz |
| Swiss National Science Foundation | 310030_192566 | Michel O Steinmetz |
| JSPS KAKENHI | 16K07328/17H03668 | Shinji Kamimura |
| National Natural Science Foundation of China | 30930108 | Wei-Shuo Fang |
| Chinese Academy of Medical Sciences | 2016-I2M-1-010 | Wei-Shuo Fang |
| Ministerio de Ciencia e Innovación | PID2019-104070RB-C22 (AEI/10.13039/501100011033) | Federico Gago |

The funders had no role in study design, data collection and interpretation, or the decision to submit the work for publication.

## Author contributions

Andrea E Prota, Conceptualization, Investigation, Visualization, Writing - original draft, Writing - review and editing; Daniel Lucena-Agell, Yuntao Ma, Juan Estevez-Gallego, Shuo Li, Katja Bargsten, Fernando Josa-Prado, Natacha Gaillard, Shinji Kamimura, Tobias Mühlethaler, Investigation; Karl-Heinz Altmann, Conceptualization; Federico Gago, Investigation, Visualization; Maria A Oliva, Investigation, Visualization, Writing - review and editing; Michel O Steinmetz, Conceptualization, Writing - review and editing; Wei-Shuo Fang, Conceptualization, Investigation, Writing - review and editing; J Fernando Díaz, Conceptualization, Investigation, Writing - original draft, Project administration, Writing - review and editing

## Author ORCIDs

Andrea E Prota http://orcid.org/0000-0003-0875-5339
Daniel Lucena-Agell http://orcid.org/0000-0001-7314-8696
Fernando Josa-Prado http://orcid.org/0000-0002-6162-3231
Federico Gago http://orcid.org/0000-0002-3071-4878
Maria A Oliva http://orcid.org/0000-0002-2215-4639
J Fernando Díaz http://orcid.org/0000-0003-2743-3319

## Decision letter and Author response

Decision letter https://doi.org/10.7554/eLife.84791.sa1
Author response https://doi.org/10.7554/eLife.84791.sa2

# Additional files

## Supplementary files
• MDAR checklist

## Data availability

Requests for samples of the newly compounds synthesized should be addressed to W-S. F. Coordinates and structure factors have been deposited at the Protein Data Bank (http://www.rcsb.org/) under accession numbers PDB: 8BDE ($T_2R$-TTL-**BacIII**), 8BDF ($T_2R$-TTL-**2a**), and 8BDG ($T_2R$-TTL-**2b**).

The following datasets were generated:

| Author(s) | Year | Dataset title | Dataset URL | Database and Identifier |
|-----------|------|---------------|-------------|-------------------------|
| Prota AE, Steinmetz MO | 2022 | TR2-TTL- Baccatin III complex | https://www.rcsb.org/structure/8BDE | RCSB Protein Data Bank, 8BDE |
| Prota AE, Steinmetz MO | 2022 | T2R-TTL-2a complex | https://www.rcsb.org/structure/8BDF | RCSB Protein Data Bank, 8BDF |

*Continued on next page*

*Continued*

| Author(s) | Year | Dataset title | Dataset URL | Database and Identifier |
|---|---|---|---|---|
| Prota AE, Steinmetz MO | 2022 | T2R-TTL-2b complex | https://www.rcsb.org/structure/8BDG | RCSB Protein Data Bank, 8BDG |

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
