## [Editor Report]

Here Prota et al. compare the action of the microtubule-stabilizing agent, taxol, with that of closely-related analogues and, as a result, successfully dissect the interactions and roles of different regions of the taxol molecule. The overall story is solid, providing new molecular insights, including defining and separating the lattice expansion effect from the lattice stabilization effect upon taxane binding. This important work will be of interest to the microtubule cytoskeleton and structural biology communities.

---

## [Decision Letter]

**Decision letter after peer review:**

Thank you for submitting the paper "Structural insight into the stabilization of microtubules by taxanes" for consideration by *eLife*. Your article has been reviewed by 2 peer reviewers, and the evaluation has been overseen by a Reviewing Editor and a Senior Editor.

*Reviewer #1 (Recommendations for the authors):*

This is careful, valuable work. I am enthusiastic that the authors have experimentally separated the lattice expansion effect of taxane binding from the lattice stabilisation effect, using baccatin III, an analogue that expands the lattice but does not stabilise it. Compared to an earlier iteration (BiorXiv) this no longer reports NMR data on taxane binding in solution and the result is a clearer story. The authors combine crystallography on the binding of taxanes to unassembled tubulin with work on taxane binding to the built lattice (via cryoEM and solution fibre diffraction from aligned microtubules) and molecular dynamics simulations.

The work compares the action of taxol with that of 4 closely-similar analogues and as a result successfully dissects the interactions and roles of different regions of the taxol molecule. The lattice binding of taxol is known to be spectacularly stronger than its free tubulin binding and the study determines definite mechanistic causes for this difference, especially that assembly reconfigures the β M-loop, which otherwise largely occludes the taxane site. The essential role of the taxol C13 side chain in lattice stabilisation is revealed.

Molecular dynamics simulations are used to investigate the conformational dynamics. These data support the idea that the M-loop configuration controls taxol binding and taxol-dependent lattice stabilisation, and recapitulate taxane-induced lattice expansion. The simulations also suggest that taxol binding to protofilaments does not straighten them. This contradicts a well-known result in the literature, but the apparent conflict may be less stark than it seems, since the earlier publication reported that curved protofilaments and rings formed gradually in the presence of taxol.

– Baccatin III expands the lattice but does not stabilise. Unless I misunderstand, clear separation of the stabilisation and expansion mechanisms of taxol has not previously been achieved. Please consider mentioning in the abstract.

– X-ray fibre diffraction is used to measure the effects of analogues on lattice spacing, establishing in wet experiments in solution that all analogues expand the GDP lattice, by an amount similar to GMPCPP, independent of whether they stabilise the lattice. This is important and not currently mentioned in the Abstract.

– lines 538-547 the 'In conclusion' statement concisely lists the major claims of this really nice work. Consider using this material in the abstract?

– line 252 'the interacting part of the M-loop' confusing. I think this refers to the regions on the neighbouring β tubulin that bind the M-loop.

– line 254-259 also unclear. This paragraph reads like a series of notes to self instead of an address to the reader. 'Should have' is confusing. 'Crosstalk via the nucleotide' means an allosteric effect on the conformation of the distant nucleotide site?

– lines 309-325 this paragraph seems to be restating the problems (.. 'mechanistic questions'), addressed by the entire paper. It doesn't feel necessary. The next para states the problems arising out of the earlier results and sets out to address them by molecular dynamics, which makes more sense.

– p374 "laxer" binding – this caused me to stumble and ponder. Perhaps 'less stringent requirements' or similar.

­– line 512 'differences in lateral contacts' is ambiguous.

– line 546 '.. differences in lateral contacts of MTs..' (in shear flow) – clarify this means lateral contacts between protofilaments and not between microtubules in the shear flow.

– lines 194-202 I found this discussion strange, it attempts to draw together a picture of taxol binding to curved tubulin in solution having determined mechanistic reasons why this binding is heavily unfavourable. The emphasis is 'we found a way to do it' using analogues that do bind. Whereas from my perspective the important point is that the work determines mechanistic reasons underlying the observation that taxol is bad at binding to free tubulin. I may be missing something key, (perhaps there are good reasons to think this pathway, slow and unfavourable binding to free tubulin, is biologically important?); if so the reader needs to have the (?potential) significance and importance of this pathway briefly explained. This is mentioned later (line 322 et seq).

– Please clarify which tubulin isotypes are used in the simulations, and that mixed isotype brain tubulin is used for the crystallography / fibre diffraction work

– Please provide assurance that the duration of the molecular dynamics simulations was sufficient to sample adequately the conformational dynamics.

– In the molecular dynamics work the results are said to reproduce lattice expansion but I could not find a quantitative comparison with the fibre diffraction measurements.

*Reviewer #2 (Recommendations for the authors):*

This work was performed as best as was capable within the currently available methods. When possible, I would use the PDB ID codes (even the new ones 8BDE, etc.) in the manuscript as well as the figure captions to make the paper easier to follow.

If this was an EM structure, I would be less inclined to support it. X-ray is still the gold standard even if it may not always have biological relevance. Your statement referring to 3-5 angstroms of EM resolution is generous (3.5-10 from my work in the applied setting).

The movies only ran for 2 seconds for me, you may wish to make them longer if this was not a technical problem on my end.

---

## [Author Response]

Reviewer #1 (Recommendations for the authors):This is careful, valuable work. I am enthusiastic that the authors have experimentally separated the lattice expansion effect of taxane binding from the lattice stabilisation effect, using baccatin III, an analogue that expands the lattice but does not stabilise it. Compared to an earlier iteration (BiorXiv) this no longer reports NMR data on taxane binding in solution and the result is a clearer story. The authors combine crystallography on the binding of taxanes to unassembled tubulin with work on taxane binding to the built lattice (via cryoEM and solution fibre diffraction from aligned microtubules) and molecular dynamics simulations.The work compares the action of taxol with that of 4 closely-similar analogues and as a result successfully dissects the interactions and roles of different regions of the taxol molecule. The lattice binding of taxol is known to be spectacularly stronger than its free tubulin binding and the study determines definite mechanistic causes for this difference, especially that assembly reconfigures the β M-loop, which otherwise largely occludes the taxane site. The essential role of the taxol C13 side chain in lattice stabilisation is revealed.Molecular dynamics simulations are used to investigate the conformational dynamics. These data support the idea that the M-loop configuration controls taxol binding and taxol-dependent lattice stabilisation, and recapitulate taxane-induced lattice expansion. The simulations also suggest that taxol binding to protofilaments does not straighten them. This contradicts a well-known result in the literature, but the apparent conflict may be less stark than it seems, since the earlier publication reported that curved protofilaments and rings formed gradually in the presence of taxol.– Baccatin III expands the lattice but does not stabilise. Unless I misunderstand, clear separation of the stabilisation and expansion mechanisms of taxol has not previously been achieved. Please consider mentioning in the abstract.

The observation is now mentioned in the abstract, as suggested.

– X-ray fibre diffraction is used to measure the effects of analogues on lattice spacing, establishing in wet experiments in solution that all analogues expand the GDP lattice, by an amount similar to GMPCPP, independent of whether they stabilise the lattice. This is important and not currently mentioned in the Abstract.

The observation is now mentioned in the abstract, as suggested.

– lines 538-547 the 'In conclusion' statement concisely lists the major claims of this really nice work. Consider using this material in the abstract?

We have used part of these materials in the abstract, as suggested.

– line 252 'the interacting part of the M-loop' confusing. I think this refers to the regions on the neighbouring β tubulin that bind the M-loop.

Yes, it refers to the secondary structural elements of β tubulin that interact with the M-loop of the neighboring protofilament. We have corrected the text accordingly.

– line 254-259 also unclear. This paragraph reads like a series of notes to self instead of an address to the reader. 'Should have' is confusing. 'Crosstalk via the nucleotide' means an allosteric effect on the conformation of the distant nucleotide site?

We have now made it more clear, that the cross-talk to the nucleotide, and indirect to the T5-loop happens through a reduction of dynamicity of helix H7, caused by taxanes binding to the site.

Now lines 295-299 and 305-306.

– lines 309-325 this paragraph seems to be restating the problems (.. 'mechanistic questions'), addressed by the entire paper. It doesn't feel necessary. The next para states the problems arising out of the earlier results and sets out to address them by molecular dynamics, which makes more sense.

We have deleted the paragraph following the reviewer’s suggestion. Relevant references were integrated into the remaining text.

– p374 "laxer" binding – this caused me to stumble and ponder. Perhaps 'less stringent requirements' or similar.

We have modified the sentence according to the suggestion

­– line 512 'differences in lateral contacts' is ambiguous.– line 546 '.. differences in lateral contacts of MTs..' (in shear flow) – clarify this means lateral contacts between protofilaments and not between microtubules in the shear flow.

We have changed the word lateral for interprotofilament

– lines 194-202 I found this discussion strange, it attempts to draw together a picture of taxol binding to curved tubulin in solution having determined mechanistic reasons why this binding is heavily unfavourable. The emphasis is 'we found a way to do it' using analogues that do bind. Whereas from my perspective the important point is that the work determines mechanistic reasons underlying the observation that taxol is bad at binding to free tubulin. I may be missing something key, (perhaps there are good reasons to think this pathway, slow and unfavourable binding to free tubulin, is biologically important?); if so the reader needs to have the (?potential) significance and importance of this pathway briefly explained. This is mentioned later (line 322 et seq).

We thank the reviewer for this comment. We have now rephrased this paragraph and highlight the structural details that support why paclitaxel preferentially binds straight tubulin. We also included a sentence to highlight the significance of these findings.

Now lines 210-221 and 236-240.

– Please clarify which tubulin isotypes are used in the simulations,

a1b2 isotype as found in PDB ID 5SYG. This is now stated in the M&M section.

and that mixed isotype brain tubulin is used for the crystallography / fibre diffraction work.

This is now described in the M&M section as requested.

– Please provide assurance that the duration of the molecular dynamics simulations was sufficient to sample adequately the conformational dynamics.

300-1000 ns is about the upper limit of current MD simulations for systems as large as those studied here. In our view, the lack of drifting to higher rmsd or distance values in Figures 8 and 9, attest to an adequate sampling of conformational space.

– In the molecular dynamics work the results are said to reproduce lattice expansion but I could not find a quantitative comparison with the fibre diffraction measurements.

The molecular dynamics experiments shown an expansion of roughly 2 Å of the ligated vs unligated lattice (this is now mentioned in the text line 427 while the outcome of the simulations is presented in Figure 9E)

Reviewer #2 (Recommendations for the authors):This work was performed as best as was capable within the currently available methods. When possible, I would use the PDB ID codes (even the new ones 8BDE, etc.) in the manuscript as well as the figure captions to make the paper easier to follow.

We have followed the suggestion and incorporated the PDB ID codes when possible.

If this was an EM structure, I would be less inclined to support it. X-ray is still the gold standard even if it may not always have biological relevance. Your statement referring to 3-5 angstroms of EM resolution is generous (3.5-10 from my work in the applied setting).

We have changed the sentence accordingly.

The movies only ran for 2 seconds for me, you may wish to make them longer if this was not a technical problem on my end.

We saved the movies as looped version, but seems as the loop setting is not saved. It’s just a technical thing. The movies can simply be looped upon opening in Quicktime (View-> Loop) or other video players.